# TopoGaussian: Inferring Internal Topology Structures from Visual Clues

**Xiaoyu Xiong**[1], **Changyu Hu**[1], **Chunru Lin**[2], **Pingchuan Ma**[3], **Chuang Gan**[2], **Tao Du**[1][4]
[1] Tsinghua University    [2] University of Massachusetts Amherst
[3] Massachusetts Institute of Technology    [4] Shanghai Qi Zhi Institute

## Abstract

We present TopoGaussian,[1] a holistic, particle-based pipeline for inferring the interior structure of an opaque object from easily accessible photos and videos as input. Traditional mesh-based approaches require tedious and error-prone mesh filling and fixing process, while typically output rough boundary surface. Our pipeline combines Gaussian Splatting with a novel, versatile particle-based differentiable simulator that simultaneously accommodates constitutive model, actuator, and collision, without interference with mesh. Based on the gradients from this simulator, we provide flexible choice of topology representation for optimization, including particle, neural implicit surface, and quadratic surface. The resultant pipeline takes easily accessible photos and videos as input and outputs the topology that matches the physical characteristics of the input. We demonstrate the efficacy of our pipeline on a synthetic dataset and four real-world tasks with 3D-printed prototypes. Compared with existing mesh-based method, our pipeline is 5.26x faster on average with improved shape quality. These results highlight the potential of our pipeline in 3D vision, soft robotics, and manufacturing applications.

## 1 Introduction

The idea of seeing through the surfaces of an opaque object is an intriguing concept contemplated by scientists, engineers, and science-fiction writers. The capability of inferring the internal topological structure of an object from its surface information unlocks many creative applications, e.g., 3D model design and industrial manufacturing. Traditional approaches from scientific or engineering disciplines either require intrusive probing sensors or operate on expensive computed tomography (CT) or magnetic resonance imaging (MRI) equipment, underscoring the challenge of this problem. Recent developments in 3D vision provide powerful tools for reconstructing 3D scenes with easily accessible, non-intrusive visual input only, e.g., NeRF (Wang et al., 2021) or Gaussian Splatting (Kerbl et al., 2023) with photos or videos as input. However, these methods only provide visually plausible *surface* information of objects with no or hallucinated interior topology structure.

This paper focuses on tackling the following problem: Given the motion (images or videos) of the object, find *one* interior topological design that can explain this motion physically. We develop a holistic pipeline for inferring the interior topology of an opaque object in order to recover its major physical characteristics with easily accessible visual input (photos and videos) only. Our pipeline, TopoGaussian, begins with collecting multi-view images and uses Gaussian splatting to obtain a point cloud encoding the object's exterior surface shape and appearance (Kerbl et al., 2023). Next, we construct a volumetric point cloud by filling interior points (Xie et al., 2023), and we use a topology representation to attach the physical parameters to each point. Here we provide three options to encode the topology structure. One option depicts the boundary directly based on a topology parameter of each particle, and map this parameter to its physical attribute, inspired by standard topology optimization methods on voxels (Sigmund, 2001a) and particles (Li et al., 2021); the other two options leverages an parametric boundary surface based on neural network (Park et al., 2019) and quadrics (Hafner et al., 2024) respectively to represent the topology structure, and assign physical attributes based on this boundary. To accommodate the particle representation, we develop

---

[1] https://topo-gaussian.github.io/TopoGaussian/

a novel, versatile particle-based differentiable simulator with simultaneous support for rigid and soft objects with different topology structure. Finally, we compare the simulation results and the object's dynamic motion in a video and optimize the loss defined on the difference between them. With access to the simulation gradient, we run gradient-based methods to optimize the parameters in topology representation and obtain an interior topology structure that leads to similar motions established in the input video. Our pipeline presents a mesh-free representation for flexible and expressive modeling, simulation, and optimization of opaque objects gracefully unified in a holistic framework of particles only. This avoids tedious and error-prone mesh fixing and filling, and outputs a more smooth and 3D-print-friendly result which can be used in manufacturing. While traditional, intrusive methods can output an accurate structure that we cannot match, they require expensive and complex equipment. Our non-intrusive pipeline only uses easily accessible images and videos as input with low cost, exhibiting potential in the applications that focus on general outside behaviours (like generative models).

We design several synthetic tasks and four real-world tasks to demonstrate the efficacy of our pipeline. We present a synthetic dataset and perform benchmark experiments comparing our method with two strong mesh-based baselines PGSR (Chen et al., 2024) and Gaussian Surfels (Dai et al., 2024), based on the processing time and reconstruction quality (3D printability). Our method is 5.26x faster and 4.81x faster than PGSR in two different settings, respectively, and 1.28x faster than Gaussian Surfels, where Gaussian Surfels fails to output visually correct result in two examples. Meanwhile, our method exhibits significantly higher quality of reconstructed topology structure than the baselines, with 2.33x, 2.5x and 2.55x respectively. Then we 3D print one example to verify the feasibility in real manufacturing. Moreover, we extend our method to more synthetic validations, four real world validations, and several ablation studies, which show the ability of our pipeline to handle input under various conditions.

In summary, this work makes the following contributions:

- We develop a novel, non-intrusive pipeline for reconstructing motion with a physically correct internal topology structure of solid objects from visual inputs only, i.e., multi-view images and videos of motion.
- We present a particle-based differentiable simulation compatible with three flexible topology representations including particle, neural implicit surface, and quadratic surface. Together with Gaussian splatting, our simulation and optimization methods lead to a completely meshless pipeline.
- We demonstrate the quality and efficiency of our pipeline quantitatively against two strong baselines, and we further validate the generalizability of our method on a wide range of synthetic and real-world scenarios with diverse yet complex physics.

## 2 RELATED WORK

**3D Scene Reconstruction**   Reconstructing 3D scenes from visual inputs is a fundamental problem extensively studied in computer vision for decades. Recent representative methods for this problem include NeRF (Wang et al., 2021) and Gaussian Splatting (Kerbl et al., 2023). NeRF bases its scene geometry representation on implicit surfaces learnable from multi-view images. Several improvements are conducted in its speed, quality, and applicability in dynamic and deformable objects (Barron et al., 2021; Deng et al., 2022; Pumarola et al., 2021a; Martin-Brualla et al., 2021). Gaussian Splatting (Kerbl et al., 2023) is optimization-based and built upon a point-cloud representation, and follow-up studies have proposed various improvements in its efficiency and reconstruction quality (Chen et al., 2023; Matsuki et al., 2023; Tang et al., 2023; Wu et al., 2023; Yi et al., 2023). In particular, Xie et al. (2023) showcases novel synthesis of motions from editing a dynamic model recovered from Gaussian Splatting, and Guo et al. (2024) reconstructs physically compatible objects using a quasi-static simulation. However, these works primarily focus on reconstructing visually plausible *surface* information of an object without inferring a dynamically plausible *volumetric* structure.

**Differentiable Simulation**   Differentiable simulation extends traditional simulation with gradients and revives gradient-based optimization in graphics, learning, and robotics applications (Yu et al., 2023; Werling et al., 2021; Xu et al., 2021; Lin et al., 2021; Qiao et al., 2021; Li et al., 2022). Previous studies have developed differentiable simulators for various physical systems including rigid

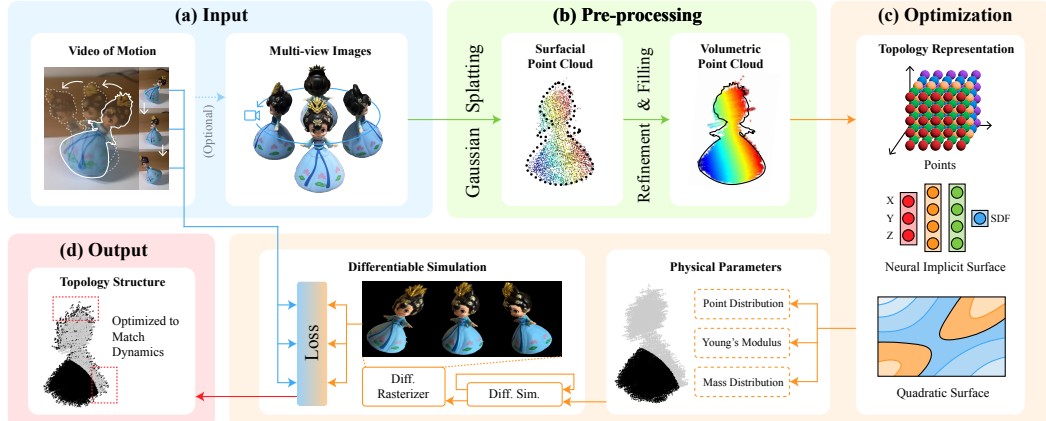

Figure 1: *Pipeline overview* (Sec. 3). Our pipeline takes as input multi-view photos of an opaque object and its video of motion. We run Gaussian splatting on the multi-view images to obtain a point cloud characterizing its surface geometry and appearance. Next, we refine and fill in internal points to obtain a volumetric point cloud and use our topology representation (with three flexible choices) to attach physical parameters on it. We then simulate the volumetric point cloud with our particle-based differentiable simulator, which compares its simulated motion with that in a reference image or video and backpropagates the gradient of the motion difference to the topology representation. Finally, we perform the optimization algorithm based on the gradient from the simulator and obtain the resulting structure that matches the input motion.

objects (Geilinger et al., 2020; Wang et al., 2019), deformable solids (Liu et al., 2023; Gjoka et al., 2022), cloths (Yu et al., 2023), and fluids (Li et al., 2024). Many of these differentiable simulators are mesh- or finite-element-based due to their benefits in handling irregular shapes with an explicit sharp boundary representation. However, these methods require high-quality mesh discretization, which can be computationally expensive to obtain. In contrast to these works, we focus on a purely particle-based differentiable simulator supporting shapes with different topology structures in accommodation with Gaussian splatting representation. This avoids tedious mesh fixing and filling by native support for a particle-only pipeline.

**Topology Optimization**   Topology optimization is an optimization-based structure design technique originating from structural and mechanical engineering (Sigmund, 2001a; Eschenauer & Olhoff, 2001; Wu et al., 2021; Deaton & Grandhi, 2014). Traditional topology optimization techniques consider a density-field representation of a structural geometry on a regular grid and optimizes independent decision variables at each voxel for optimal structural or mechanical performance, e.g., minimum compliance (Sigmund, 2001a; Yang & Chen, 1996; Liu et al., 2018). Previous studies have also explored the idea with different geometric representations, e.g., level sets (Wang et al., 2003) and particles (Li et al., 2021). Recent years, neural implicit surface method has been proposed (Zehnder et al., 2021), which encodes the shape with a deep neural network (Park et al., 2019). Moreover, for rigid bodies, Hafner et al. (2024) has proved that quadratic surfaces are sufficient for topology optimization problems that only involve rigid physical characteristics. Existing topology optimization works primarily focus on applications in engineering disciplines, e.g., structural and mechanical analysis, aeronautics, and architecture (Jensen & Sigmund, 2011; Zhu et al., 2016; Dühring et al., 2008), while our pipeline extends topology optimization technique to the applications in 3D computer vision and robotics.

## 3 METHOD OVERVIEW

The overarching goal of our pipeline is to find the internal topology structure of an opaque object to match its physically simulated motion with its dynamic motion from a reference input. Our pipeline contains four steps: volumetric shape generation (Sec. 4), which takes as input multi-view photos of a possibly hollowed opaque object and generates a densely sampled volumetric point cloud; topology representation (Sec. 5), which encodes the topology structure with flexible topology

representations and attaches a physical parameter (e.g. density or Young's modulus) to each point in the point cloud; simulation (Sec. 6), which evaluates the dynamic motion of the point cloud for a given interior topology design configured by each point's physical parameter; optimization (Sec. 7), which runs gradient-based optimization to adjust the parameters in the topology representation based on the loss of the simulated motion and the reference motion, and output a shape that replicates the dynamic motion of the object in a reference video. Fig. 1 gives an overview of the pipeline.

## 4 VOLUMETRIC SHAPE GENERATION

We begin by collecting multi-view photos of the object and running Gaussian Splatting due to its efficient and high-quality reconstruction of 3D objects from images. The output point cloud from Gaussian Splatting gives visually and geometrically plausible surface information about the object based on particles. This point-cloud representation of surface information serves as a basis towards the optimization of interior topology structure. Following PhysGaussian (Xie et al., 2023), we refine and fill in points to the output point cloud from Gaussian Splatting, leading to a clean and densely sampled volumetric point cloud $\{(\boldsymbol{X}_i, V_i)\}_{i=1}^N$ where $N$ denotes the number of points, $\boldsymbol{X}_i$ their 3D locations, and $V_i$ the volume associated with $\boldsymbol{X}_i$. The point cloud can be viewed as a particle-based discretization of the solid object.

## 5 TOPOLOGY REPRESENTATION

Now we have a volumetric point-cloud, and we need to represent different interior topology within it, including pneumatic hollows and bone-muscle structures. Based on the point-cloud and previous work (Li et al., 2021; Park et al., 2019; Hafner et al., 2024), we provide two topology representations, point-based and surface-based, explained in Sec. 5.1 and Sec. 5.2 respectively.

### 5.1 POINT REPRESENTATION

For an internal topology structure, we use an indicator function $r(\boldsymbol{x})$ to define. For example, in a pneumatic actuator, we use $r(\boldsymbol{x}) = 1$ to indicate the hollow part and $r(\boldsymbol{x}) = 0$ to indicate the solid part. Then we discretize this indicator function to the point-cloud by assigning a value to each point that controls the nearby area of this point. While strict binary variable is discrete and not compatible with gradient-based optimizer (Sigmund, 2001b), we consider assigning a decision variable to each point and using a sigmoid function to map it to the binary variable. Then the topology structure is represented by this decision variable.

### 5.2 SURFACE REPRESENTATION

Point representation exhibits highly flexible topology representation, but it is hard to encode specific geometric prior information into it. Based on this, our pipeline further provides two topology representations based on a parametric surface. We use a signed distance function (SDF) $s(\boldsymbol{x})$ to encode the boundary surface of the object, where $\boldsymbol{x}$ is any position inside the object, and $s(\boldsymbol{x})$ outputs the signed distance from this position to the boundary surface. Then the surface is the zero level set ($s(\boldsymbol{x}) = 0$) of the SDF (Osher et al., 2004). Similarly, we discretize the SDF to the point-cloud, and use a sigmoid function to map the signed distance to a binary indicator function. We provide two concrete SDF representations to parameterize the surface: neural networks (Park et al., 2019) and quadrics (Hafner et al., 2024). In particular, quadrics are proved to be sufficient for topology optimization on specific rigid-body dynamic tasks (Hafner et al., 2024) with only 10 parameters.

## 6 SIMULATION

With the volumetric point-cloud containing physical parameters, the next step in our pipeline involves simulating the dynamic motion which we can draw a comparison with the input motion and perform optimization. Our work focuses on rigid and soft solid objects, both of which have mature simulation solutions based on (finite-element) mesh representations Sifakis & Barbic (2012a); Sin et al. (2013); Lee et al. (2018). However, as we pointed out, obtaining high-quality mesh is

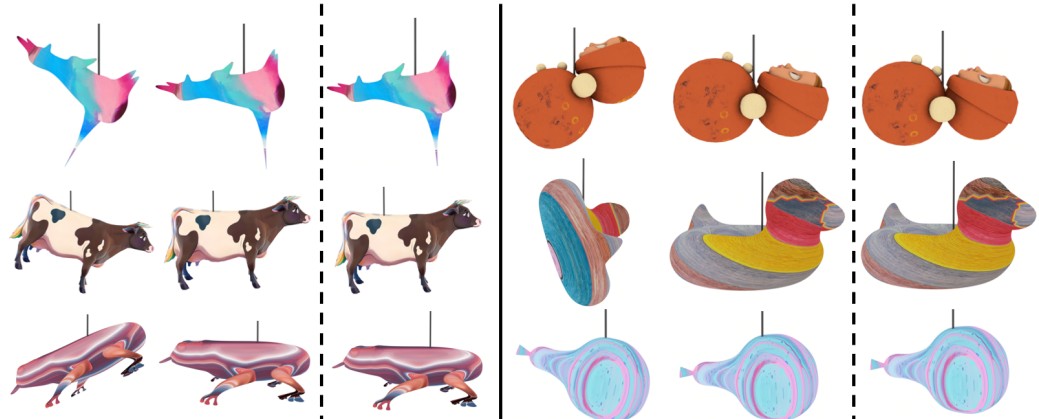

Figure 2: *Six rigid experiments* (Sec. 8.2). For each experiment, left: initial balancing position of the object with fully solid topology structure; middle: final balancing position of the object with the result from our pipeline; right: optimization target.

non-trivial, requiring error-prone and tedious efforts, while the output from Gaussian Splatting is point-cloud. Therefore, our pipeline proposes to consider a fully particle-based simulator instead and aims to build a completely mesh-free pipeline. We note that previous works like PhysGaussian (Xie et al., 2023) have discussed a similar motivation behind a fully particle-based pipeline, but we are faced with extra technical challenges from representing the internal geometric structure in the particle system. We discuss the detail of our simulator in the following parts: constitutive models, actuation model, collision model, time integration and differentiability.

## 6.1 CONSTITUTIVE MODELS

**Rigid Bodies** The dynamic motion of rigid bodies is governed by the Newton-Euler equation (Lanczos, 2012; Liu & Jain, 2012) on six DoFs. Specifically, the governing equation requires access to the mass $m$, the center of mass $c$, and the moment of inertia $\boldsymbol{I}$. From the indicator function $r(\boldsymbol{x})$ defined before, we can calculate the mass of the particle near $i$ from $m_i = \rho V_i r(\boldsymbol{x}_i)$. Then the dynamic properties can be computed from $\{(\boldsymbol{x}_i, m_i)\}$ in a straightforward manner. For example, the mass $m = \sum_i m_i$, and the center of mass $c$ can be computed as

$$\boldsymbol{c} = \frac{\sum_i m_i \boldsymbol{x}_i}{m}. \tag{1}$$

Similarly, the moment of inertia can be computed from

$$\boldsymbol{I} = \sum_i -m_i [\boldsymbol{x}_i - \boldsymbol{c}][\boldsymbol{x}_i - \boldsymbol{c}], \tag{2}$$

where $[\cdot]$ maps a vector to its cross-product (skew-symmetric) matrix.

**Soft Bodies** We consider simulating a soft body made of hyperelastic materials. The critical step is to estimate the deformation gradient $\boldsymbol{F}_i$, a $3 \times 3$ tensor characterizing the local deformation around each $\boldsymbol{X}_i$. Here, we follow the standard practice in particle-based simulation for deformable solids (Müller et al., 2005; 2022) to construct $\boldsymbol{F}_i$. More concretely, given a deformed point cloud $\{\boldsymbol{x}_i\}$ during simulation, we estimate for any point $\boldsymbol{x}_i$ its $\boldsymbol{F}_i$ with the method in Becker et al. (2009):

$$\boldsymbol{F}_i = \boldsymbol{I} + \left( \sum_j V_j (\boldsymbol{R}_i^\top (\boldsymbol{x}_j - \boldsymbol{x}_i) - (\boldsymbol{X}_j - \boldsymbol{X}_i)) \nabla W (\boldsymbol{X}_i - \boldsymbol{X}_j) \right)^T, \tag{3}$$

where $j$ loops over particles in a neighborhood of $\boldsymbol{X}_i$, and $\boldsymbol{R}_i^\top$ is a rotation transformation based on the relative poses of these neighboring particles (Becker et al., 2009). With the definition of $\boldsymbol{F}_i$ at hand, we can calculate the energy density $\Psi$ and stress tensor $\boldsymbol{P}$ based on the Saint Venant–Kirchhoff model (Sifakis & Barbic, 2012b).

Table 1: *Reconstruction quality.* The lower the better. The optimal results are **bolded**.

| Method | frog | swim-ring | cow | kangaroo | pear | tumbler |
|---|---|---|---|---|---|---|
| Ours | **0.306** | **0.242** | **0.300** | **0.388** | **0.245** | **0.305** |
| PGSR (voxel size=0.05) | 0.745 | 0.464 | 0.410 | 0.841 | 1.078 | 0.953 |
| PGSR (voxel size=0.2) | 0.781 | 0.894 | 0.801 | 0.480 | 1.027 | 0.755 |
| Gaussian Surfels | 0.851 | 0.882 | Fail | 0.976 | 1.148 | 0.963 |

Figure 3: *Comparison on optimization loss (left) and preprocessing time (right).* "Fail" on the top of the bar means that the method fails to output visually correct result in the example.

## 6.2 ACTUATION MODEL

In order to extend the applications of the previous soft-body model and design soft bodies which can perform active motion, we adopt the actuation model in soft robotics. In particular, the pneumatic chamber embedded in the soft body is typically characterized by a customized chamber shape design invisible from the soft body's exterior surface. We assume that the air pressure force induced by the pneumatic actuator is the conservative force from the volume energy (Luo et al., 2022):

$$E_a(\{\boldsymbol{x}_i\}) = \int_{\Omega_{\{\boldsymbol{x}_i\}}} p\,dv, \tag{4}$$

where $p$ denotes the air pressure. Note that the integral domain is the volumetric region of the *deformed* shape. With $\boldsymbol{F}_i$ at hand, we can approximate $E_p$ as

$$E_a \approx \sum_i p v_i = \sum_i p|\boldsymbol{F}_i|V_i, \tag{5}$$

where $|\boldsymbol{F}_i|$ represents the determinant of $\boldsymbol{F}_i$. Therefore, to support pneumatically actuated soft-body simulation, we use the conservative force derived from $E_a$ as the actuation force.

## 6.3 COLLISION MODEL

We follow the existing study on penalty-based collision simulation (Andrews et al., 2022) to build our collision model, leveraging the advantage of the penalty-based method that it properly balances accuracy, speed, and differentiability. More concretely, we add for each particle of the soft body a penalty energy density

$$E_{pi} = \frac{1}{2}k\left(\min\{\delta - d_i, 0\}\right)^2 \tag{6}$$

to discourage it from penetrating the obstacles. In Eqn. (6), $k$ is the penalty stiffness, $\delta$ is a small translation parameter to avoid penetration, and $d_i$ is the signed distance between particle $i$ and the obstacle (e.g. ground). Then, the contact force is also the conservative force derived from the penalty energy.

## 6.4 TIME INTEGRATION

With the total energy and force of the whole system, we can perform time integration to solve the motion. For rigid bodies, based on the Newton-Euler equations (Lanczos, 2012; Liu & Jain, 2012), we perform RK2 integration. For soft bodies, we use Leapfrog integration for non-actuation cases

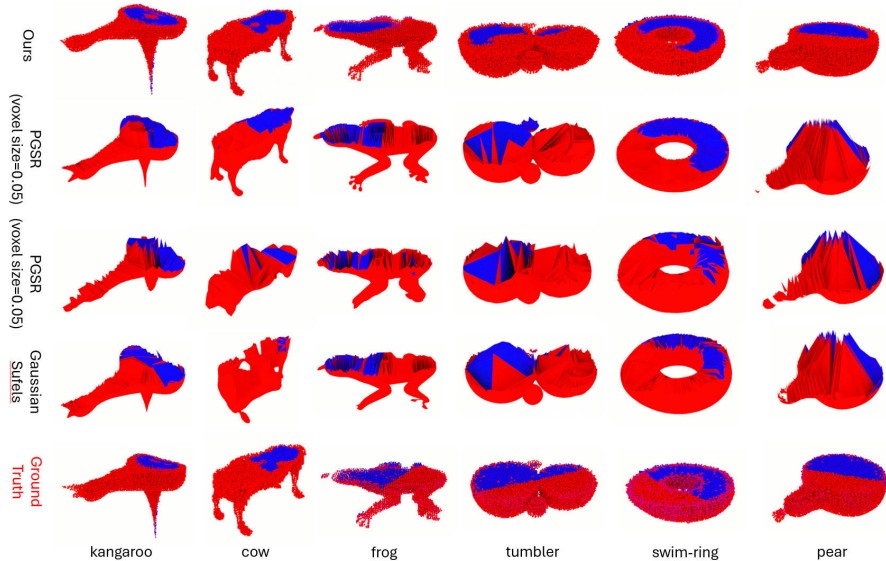

Figure 4: *Comparison between our method and mesh baselines* (Sec. 8.1). Red part represents solid part and blue part represents hollow part. Top row: our method; bottom three rows: mesh baselines.

and implicit Euler integration with incremental potential (Martin et al., 2011) in actuation cases. The details of incremental potential can be found in Appendix Sec. A.2.

## 6.5 GRADIENT COMPUTATION

We apply the chain rule and the standard adjoint equation (Du et al., 2021; Geilinger et al., 2020; Hahn et al., 2019; McNamara et al., 2004) to equip the proposed particle-based simulator with gradient information. Detailed derivations of the gradients in rigid-body and deformable-body simulators can be found in previous works, e.g., ChainQueen (Hu et al., 2019) and DiffPd (Du et al., 2021) for deformable bodies, Huang et al. (2024) for pneumatic actuators, and Andrews et al. (2022) for collision models.

## 7 OPTIMIZATION

The final step of our pipeline is the optimization based on the L2 loss from the difference between the simulation motion from the previous section and the input reference motion. (See Fig. 1.) We extract certain motion characteristics (e.g. vibration frequency, balancing pose) and define the loss based on the characteristics. The optimization variable is the parameters in the topology representation (Sec. 5). We use gradient-based optimizer (L-BFGS-B and Adam), reading the loss from the differentiable simulator (Sec. 6.5).

## 8 RESULTS

### 8.1 EXPERIMENT SETUP

**Task Description** We comprehensively evaluate our method on diverse tasks, including nineteen rigid bodies and two soft bodies with static and dynamic settings and their extension to four real world experiments. Each task takes multi-view images and the motion video of the object as input. Our goal is to find an interior topology structure whose simulated motion matches the reference motion, characterized by static/dynamic features like balancing position or frequency. Fig. 2 and Fig. 5 exhibit several examples of these tasks.

**Baselines** Our main task is to build a particle-based pipeline to find the physically plausible internal topology structure, which is a new task with few baselines. Recent PhysGaussian (Xie et al., 2023) does not consider the optimization of internal topology structure. Therefore, we build the

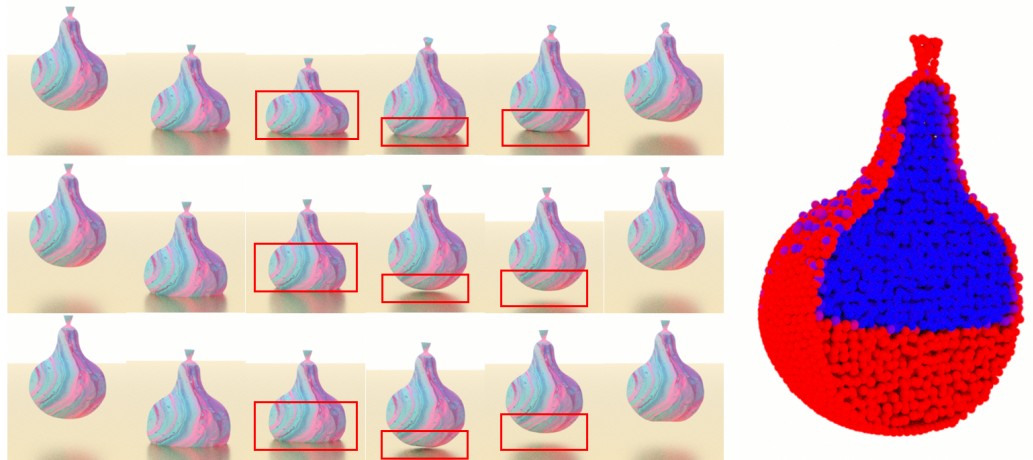

Figure 5: *One soft body expereiment* (Sec. 8.2). Left part: Top row: frames from simulated motion of a fully soft initial guess; middle row: simulated motion of our optimized result; bottom row: target motion. The red rectangles mark the major different parts. Right part: slice of the optimized topology structure with red representing soft part and blue representing hard part.

baseline based on two current mesh-based Gaussian Splatting SOTA PGSR (Chen et al., 2024) and Gaussian Surfels (Dai et al., 2024). We replace the topology representation with the output triangle surface mesh from the model, and chain it into the rest of our pipeline. By comparing the efficacy of mesh- and particle-based methods, we show the advantages of insisting on a fully particle-base method.

**Metrics**   We don't compare the output shape with the shape set in generating the target motion due to two reasons. One reason is that we only consider image/video input, and especially in rigid tasks, this comparison is ill-defined since multiple solutions can share the same motion, which means that this metric fails to measure the performance of the method. The other reason is that in practical applications, this ground truth topology structure can not be easily acquired, requiring expensive equipment like CT/MRI. Instead, we use the following three metrics to test the performances:

- **Optimization Loss**: The L2 loss between simulation motion and reference motion (Sec. 7).
- **Time**: Volumetric shape generation time from reading multi-view images input to simulation start (Sec. 4 and Sec. 5).
- **Smoothness**: The Laplacian of surface normal. This is motivated by manufacturing applications (e.g., 3D printing) which favor smooth surface.

## 8.2   MAIN RESULTS

**Rigid Body**   Rigid bodies with different shapes are hung by a string at a given position, and our goal is to optimize the topology structure of their internal cavities to remain horizontal. As this is a static equilibrium example, we first extract the center of mass $c$ of the hawk from a shape parameter configuration $s$. We stress that varying $s$ changes the interior shape, which adjusts the weight distribution of the hawk and, therefore, affects the center of mass. We define the loss as the distance between $c$ and the hanging position. Fig. 2 shows six examples and visualizes the initial, optimized, and target pose of the rigid bodies. The full results can be found in Appendix Sec. C.2.

Fig. 3 exhibits the optimization loss and volumetric shape generation time of our method and each baseline on the six experiments, and the full figure can be found in Appendix Sec. C.2. It can be observed that all these methods exhibit similar and sufficiently small final loss, meaning that they all simulate a motion close to reference. On the other hand, for volumetric shape generation time, our method is 5.26 times faster than PGSR (voxel size=0.05), 4.81 times faster than PGSR (voxel size=0.2), and 1.28 times faster than Gaussian Surfels on average. This result shows that our pipeline is more efficient owing to leveraging the flexibility of particles, thus avoiding expensive mesh processing.

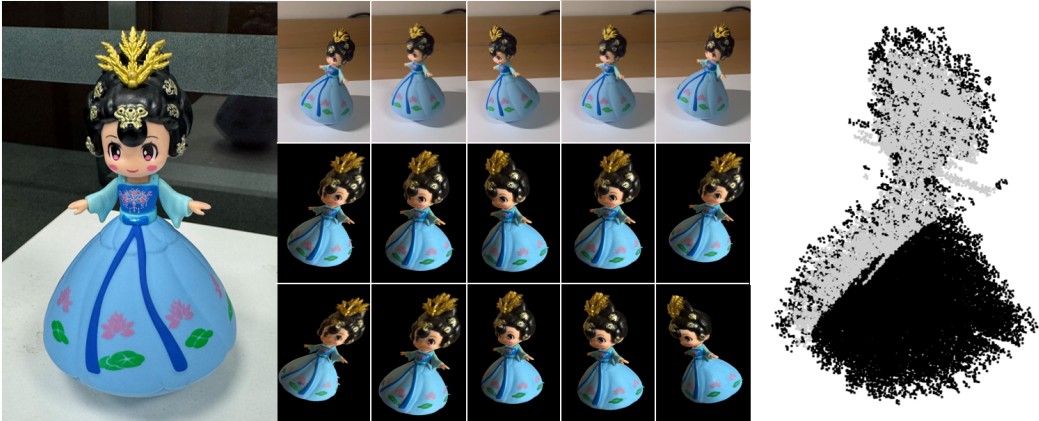

Figure 6: *Wobbly doll* (Sec. 8.3). Left: a photo of the wobbly doll. Middle: frames from a video recording of the harmonic oscillation of the doll (top), simulated motion with an interior design optimized from our pipeline (middle), simulated motion of a solid doll (bottom). Right: a cross-section visualization of the optimized interior geometry from our pipeline, where black represents solid part and grey represents hollow part.

For the quality of reconstruction, we try neural implicit surface and quadrics topology representations in our methods, and find that there is no obvious quality difference between two representations, which is corresponding to the conclusions in Hafner et al. (2024). Tab. 1 lists the quality index of our method and the baselines on the six examples, and Fig. 4 visualizes these results (complete data and visualization results in Appendix Sec. C.2). More concretely, the quality of our method is 2.33 times better than PGSR (voxel size=0.05), 2.50 times better than PGSR (voxel size=0.2), and 2.55 times better than Gaussian Surfels on average. Furthermore we observe that in the cow task, Gaussian Surfels fails to output a visually correct result. This showcases the advantages of the our particle-based pipeline,

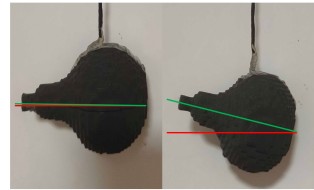

Left: optimized result; right: fully solid initial guess. Red line: target pose; green line: actual pose.

which does not require an expensive and error-prone high quality mesh division. Based on the smooth geometry from our pipeline, we 3D print the pear example to exhibit our application in manufacturing. Moreover, we perform several ablation studies shown in Appendix Sec. C.4.

**Soft Body**   We also extend our synthetic dataset to soft bodies with two experiments with point topology representation. Here we introduce one of them and the other can be found in Appendix Sec. C.2. The experiment is a soft pear bouncing on the ground, with a hard part inserted into it. Different topology structures of this hard part lead to different bouncing behaviors, while our goal is to optimize this structure to reproduce the target video. The loss is defined as the distance of the motion of several pivot points on the pear to the target.

Fig. 5 visualizes the result of the experiment. The fully soft initial guess output a motion deviating large from the target, while our optimized result can closely reproducing the target motion.

### 8.3   REAL-WORLD VALIDATIONS

Finally, we validate our pipeline on four real-world experiments. These experiments include rigid and soft bodies with static and dynamic settings. One rigid and one soft body examples are shown here, and the rest can be found in Appendix Sec. C.3

**Wobbly Doll**   The task involves inferring the interior structure of a wobbly doll, a classic rigid-body toy with oscillating motions (Fig. 6 left and top middle). The interior of a wobbly doll is typically non-empty. The goal is to infer one possible interior structure of a wobbly doll from a video recording of its harmonic oscillations (Fig. 6 top middle).

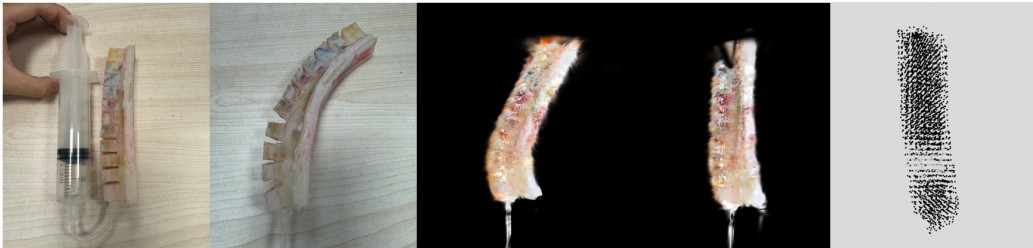

Figure 7: *Pneumatic finger* (Sec. 8.3). Left: a photo of the pneumatically actuated soft finger. Middle left: the bending finger with maximal air pressure. Middle: bending finger simulated with an optimized design. Middle right: the finger simulated with an initial guess of the chamber location; Right: the optimized chamber location where the black is solid part and the grey is chamber.

We define an objective function that penalizes the discrepancy of the oscillation period between the reference video and the simulated motion. We extract the period of oscillation as well as the maximal tilting angle $\theta$ by OpenCV. Then, we compare these reference values with the simulated value from the physical parameters and compute the loss.

Fig. 6 (right) shows our optimized interior structure of the doll, where most of the mass is clustered to one side of the bottom. The simulated motion of this optimized design (Fig. 6 middle) closely matches the maximal tilting angle and the period the recorded video (4.0Hz. See Fig. 6 top middle). In contrast, an initial guess of a fully solid doll leads to a substantially different oscillation period (2.4Hz. See Fig. 6).

**Pneumatic Finger** This experiment is inspired by a classic application in soft robotics, which showcases the ability of our pipeline in processing soft bodies with pneumatic actuators. The task is to infer the chamber design in an opaque, pneumatically actuated soft finger, a widely used model in soft robotics (Fig. 7 left). This task is substantially more challenging than previous ones as it involves deformable bodies and air pressure in a pneumatic chamber. Given a video recording of the bending motion of the soft finger with increasing air pressure, our goal is to deduce the inflatable chamber design to reproduce the final bending angle (Fig. 7 middle left). We extract reference angle from input by OpenCV, and compare it with our simulated value. Fig. 7 middle right shows that there is no obvious bending with initial guess, and a visually correct bending with our final result. The result indicate that we need to add a small chamber in the middle bottom part, which is reasonable to achieve the target bending.

## 9 CONCLUSIONS AND LIMITATIONS

We present a holistic pipeline for optimizing the interior topology structure of opaque objects from visual inputs. Our pipeline consists of Gaussian Splatting, topology optimizer with flexible representation choices, and a novel differentiable point-cloud simulator. Our whole pipeline is built upon a gracefully unified particle-based representation free from error-prone mesh extraction and processing in modeling, simulation, and optimization. We validate our pipeline on a diverse set of tasks, covering rigid bodies, soft bodies, collision and actuation with various static and dynamic scenes. Through the comparison with mesh-based methods, we show that our particle-based method has the advantages of higher speed and smoother output. Our pipeline enables promising applications in reverse engineering tasks in 3D vision and soft robotics.

Our work has several limitations. First, our pipeline currently focuses on a single object with two types of material. Second, Dynamic NeRF (Pumarola et al., 2021b) and 4D Gaussian (Wu et al., 2024) also performs 3D scene reconstruction and we do not compare with them. The main reason is that these works only consider the surface information without guarantee of a physically plausible output, meaning that they may not be able to handle the manufacturing applications like 3D printing. Despite of this, it would still be interesting to compare our approach with them. Third, current optimization, especially for neural implicit surface and soft bodies, still costs certain time and is only offline. It would be promising to accelerate them to real time and supports interactive demos.

ACKNOWLEDGMENTS

We would like to thank Tsinghua University, Shanghai Qi Zhi Institute, University of Massachusetts at Amherst, and the Massachusetts Institute of Technology for their financial support throughout this research. We also extend our gratitude to Zijian Lyu from Tsinghua University for his technical assistance and insightful discussions. Special thanks to Shuguang Li from Tsinghua University for providing the necessary hardware support that greatly facilitated our work.

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

## A  DETAILS OF ALGORITHMS

### A.1  SOFT BODY DYNAMICS

**Discretization**  We start from the discretization and interpolation of functions on the object space. For any function $f(\boldsymbol{x})$ on the object, we records its value on each particle $f_i$. Then for any position $\boldsymbol{x}$, the function value can be interpolated as

$$\langle f(\boldsymbol{x}) \rangle = \sum_i f(\boldsymbol{x}_i) W(\boldsymbol{x} - \boldsymbol{x}_i, h) \frac{m_i}{\rho_i} \tag{7}$$

where $W(\boldsymbol{r}, h)$ is the kernel function used to smooth the interpolation. $h$ is the interpolation range parameter and usually set to 1 to 2 times the average distance between particles. There are many

forms of the kernel function, while we use the version based on spline functions Monaghan & Lattanzio (1985), with the following form

$$W(\boldsymbol{r}, h) = \frac{1}{\pi h^3} \begin{cases} 1 - \frac{3}{2}q^2 + \frac{3}{4}q^3 & \text{if } 0 \le q < 1 \\ \frac{1}{4}(2 - q)^3 & \text{if } 1 \le q < 2 \\ 0 & \text{otherwise} \end{cases} \tag{8}$$

where $q = |\boldsymbol{r}|/h$. $m_i$ is the mass of particle $i$, and $\rho_i$ is the density near particle $i$, which can be computed from

$$\rho_i = \sum_j m_j W(\boldsymbol{x}_{ij}^0, h) \tag{9}$$

where $\boldsymbol{x}_{ij}^0$ is the undeformed displacement from particle $j$ to $i$.

**Geometry**  When the soft body deforms, we can write the deformation map between the deformed position $\boldsymbol{x}$ and undeformed position $\boldsymbol{x}^0$ in $\boldsymbol{x}^0 \mapsto \boldsymbol{x} = \boldsymbol{x}^0 + \boldsymbol{u}$, while the Jacobian of the deformation map can be writen in $\boldsymbol{J} = \boldsymbol{I} + \nabla \boldsymbol{u}^T$. Note that $\nabla \boldsymbol{u}^T$ can be decomposed into a scaling part and a rotation part, while only the scaling part is effective for mechanical computations. Thus, we need to extract the rotation matrix $\boldsymbol{R}_i$ for each particle based on its initial neighborhood. Define

$$\boldsymbol{A}_{pqi} = \sum_i m_i W(\boldsymbol{x}_{ij}^0, h) \left((\boldsymbol{x}_j - \boldsymbol{x}_i)(\boldsymbol{x}_j^0 - \boldsymbol{x}_i^0)^T\right) \tag{10}$$

Then $\boldsymbol{R}_i$ can be computed from the polar decomposition of $\boldsymbol{A}_{pqi}$, and

$$\nabla \boldsymbol{u}_i = \sum_j \frac{m_j}{\rho_j} \boldsymbol{u}_{ji} \nabla W\left(\boldsymbol{x}_{ji}^0, h\right)^T \tag{11}$$

where

$$\boldsymbol{u}_{ji} = \boldsymbol{R}_i^{-1}(\boldsymbol{x}_j - \boldsymbol{x}_i) - \left(\boldsymbol{x}_j^0 - \boldsymbol{x}_i^0\right) \tag{12}$$

**Mechanics**  Then we can use either non-linear Green-Saint-Venant strain tensor

$$\boldsymbol{\epsilon} = \frac{1}{2}\left(\boldsymbol{J}^T \boldsymbol{J} - \boldsymbol{I}\right) \tag{13}$$

or the linear Cauchy-Green strain tensor

$$\boldsymbol{\epsilon} = \frac{1}{2}\left(\boldsymbol{J} + \boldsymbol{J}^T\right) - \boldsymbol{I} \tag{14}$$

to calculate the stress energy of the object. Based on Saint Venant–Kirchhoff model, the stress energy can be written as

$$E = \sum_i \frac{m_i}{\rho_i} \left(\mu \boldsymbol{\epsilon} : \boldsymbol{\epsilon} + \frac{\lambda}{2}(\text{tr}(\boldsymbol{\epsilon}))^2\right) \tag{15}$$

where $\mu$ and $\lambda$ are the Lame parameters of the material. Then the stress force on each particle can be computed from the gradients of stress energy with respect to the position of each particle.

## A.2  TIME INTEGRATION

We will briefly explain the math details of the implicit time integration based on incremental potential in this section. Implicit time integration requires us to solve the following two equations

$$\boldsymbol{v}_{k+1} = \boldsymbol{v}_k + \boldsymbol{M}^{-1}\boldsymbol{f}(\boldsymbol{x}_{k+1})\Delta t \tag{16}$$

$$\boldsymbol{x}_{k+1} = \boldsymbol{x}_k + \boldsymbol{v}_{k+1}\Delta t \tag{17}$$

where $\boldsymbol{x}_k$ and $\boldsymbol{x}_{k+1}$ is the position at time step $k$ and $k + 1$, $\boldsymbol{v}_k$ and $\boldsymbol{v}_{k+1}$ is the velocity at time step $k$ and $k + 1$, $\boldsymbol{f}$ is the total force, $\Delta t$ is the size of time step, and $\boldsymbol{M}$ is the mass matrix. Notice that $\boldsymbol{f}$ is usually in a complex form, meaning that there is usually no analytical solution to this equation. To numerically solve it, we write $\boldsymbol{f}(\boldsymbol{x}) = -\nabla E(\boldsymbol{x}) + \boldsymbol{f}^{ext}$, splitting it into internal and external part. Then we have

$$\nabla E(\boldsymbol{x}_{k+1}) + \frac{1}{\Delta t^2}\boldsymbol{M}\left(\boldsymbol{x}_{k+1} - \left(\boldsymbol{x}_k + \Delta t \boldsymbol{v}_k + \Delta t^2 \boldsymbol{M}^{-1}\boldsymbol{f}^{ext}\right)\right) = 0 \tag{18}$$

If we rewrite this into

$$\nabla \left( E(\boldsymbol{x}_{k+1}) + \frac{1}{2\Delta t^2} \mathrm{tr}\left( (\boldsymbol{x}_{k+1} - \boldsymbol{y}_k)^T \boldsymbol{M}(\boldsymbol{x}_{k+1} - \boldsymbol{y}_k) \right) \right) = 0 \tag{19}$$

where $\boldsymbol{y}_k = \boldsymbol{x}_k + \Delta t \boldsymbol{v}_k + \Delta t^2 \boldsymbol{M}^{-1} \boldsymbol{f}^{ext}$. Notice that this can be converted into an optimization problem, and then we can use Newton optimizer to solve $\boldsymbol{x}_{k+1}$.

### A.3 VOLUMETRIC REPRESENTATION

We will more information in this section to explain the neural implicit network and quadratic surface representations in our pipeline.

**Neural Implicit Network**   The neural network reads the position in the Euclidean space as input, and output the SDF value of this position by encoding the internal topology into the network. More concretely, we can write it as

$$SDF(\boldsymbol{x}) = f_\theta(\boldsymbol{x}) \tag{20}$$

, where $\theta$ is the parameter representing the internal topology.

**Quadratic Surface**   Based on the conclusion from (Hafner et al., 2024), we can use a quadratic function to parametrize the internal topology of a rigid body if the problem only involves the rigid dynamic characteristics of the object. Therefore, the SDF can be written as

$$SDF(\boldsymbol{x}) = \boldsymbol{x}^T \boldsymbol{A}\boldsymbol{x} + \boldsymbol{b}^T \boldsymbol{x} + c \tag{21}$$

Considering the symmetry, there will be only 10 degrees of freedom in 3D cases, which can significantly simply the problem.

## B  DETAILS OF EXPERIMENTS

### B.1  IMPLEMENTATION DETAILS

We implemented the core steps in our pipeline in C++ and Python. We parallelized the differentiable simulator with both CUDA on GPUs and OpenMP on CPUs, combining with Nvidia Warp. We used Open3D for point cloud processing and OpenCV for reference video processing, and PyTorch for the implementation of neural implicit surface. We evaluated our pipeline on the tasks below on a server with an AMD EPYC 9754 128-Core CPU, 12× DDR5 4800 16GB (384GB in total) RAM, and 1 × NVIDIA RTX 4090 24G GPU. We used Open3D to perform point cloud processing, OpenCV for video processing, and L-BFGS-B and Adam for optimization.

### B.2  HYPER-PARAMETERS

**Neural Implicit Surface**   We implement this neural network with eight fully connected layers, each of which is applied with dropouts. All internal layers have size $512 \times 512$, and are connected with ReLU non-linearities. The network is trained with weight-normalization upon the gradients from the differentiable simulator. We train for 10000 epochs with learning rate 3e-6.

**Simulation**   The basic simulation parameters are listed in Tab. 2, while the actuation pressure of the soft hand is set to be 5e5 Pa.

## C  ADDITIONAL RESULTS OF EXPERIMENTS

### C.1  FULL GALLERY OF SYNTHETIC DATASET

We visualize a full gallery of our synthetic dataset in Fig. 8. In addition to the complex geometries, we also attach heterogeneous textures on the shapes to (1) match the case in real world and (2) alleviate the ambiguity for multi-view reconstruction.

Table 2: *Simulation parameters*. N/A means that the experiment does not involve this parameter.

| Experiment | Time step (s) | Density (kg/m$^3$) | Young's modulus (Pa) | Poisson's ratio |
|---|---|---|---|---|
| Rigid | 5e-3 | 1e3 | N/A | N/A |
| Soft | 5e-4 | 1e3 | 1.5e5 (softer); 3e7 (stiffer) | 0.4 |
| Rigid (real) | 5e-3 | 5e3 | N/A | N/A |
| Soft (real) | 5e-4 | 5e3 | 1e7 | 0.49 |

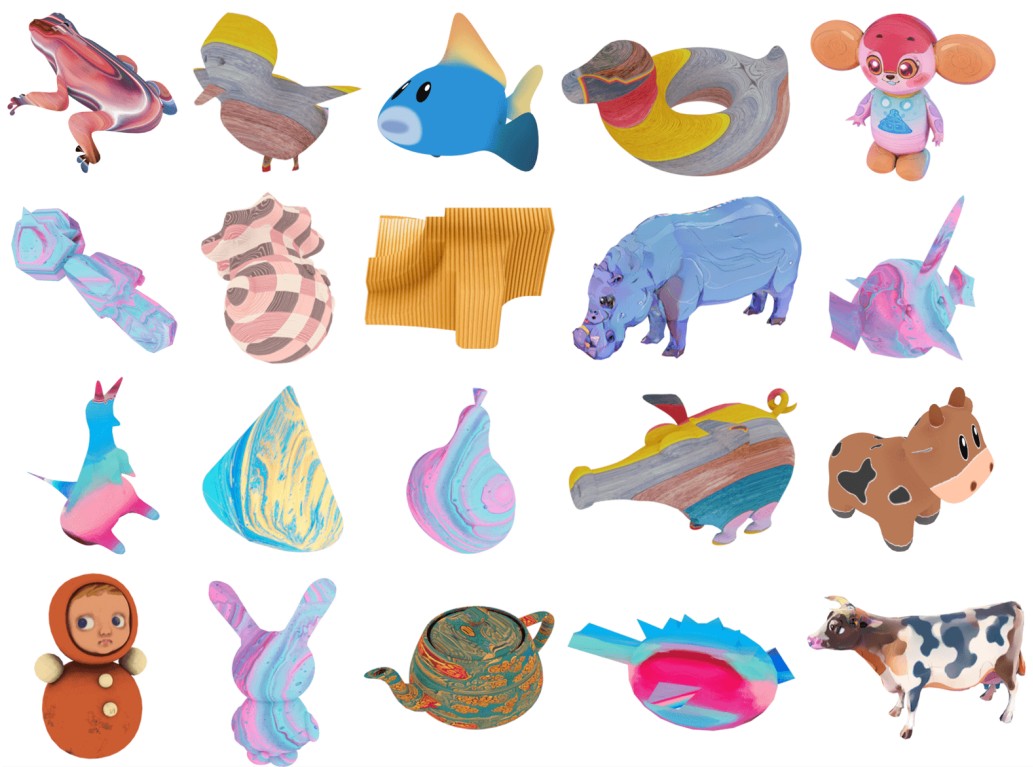

Figure 8: *Full gallery of our dataset*. We create a diverse set of different geometries with complex textures to evaluate our method.

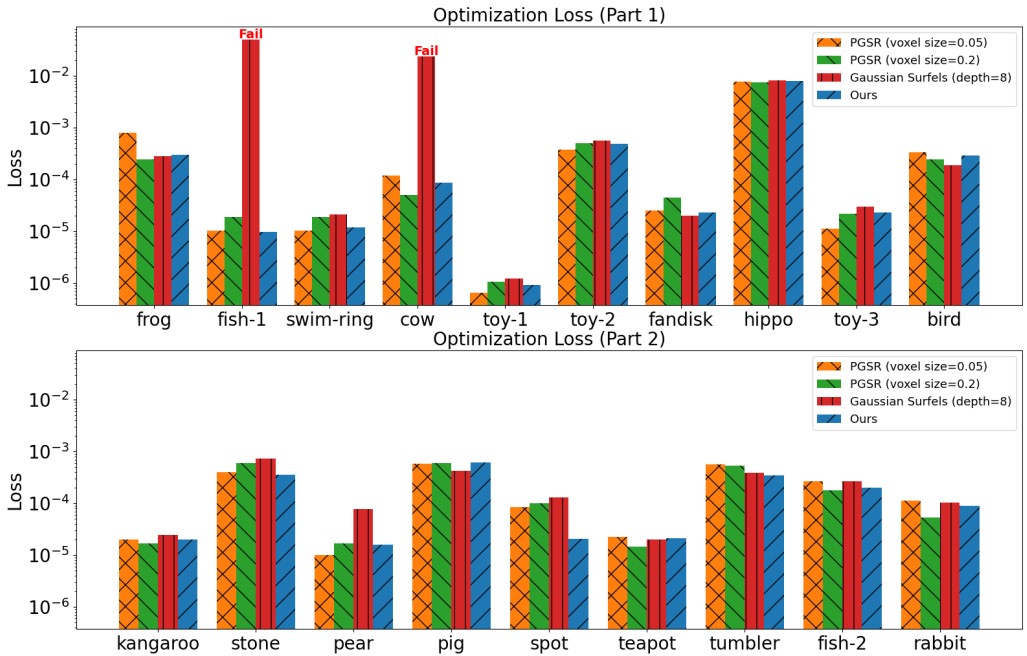

Figure 9: Full data of the optimization loss of our method and the baselines, where "fail" on the top means that the method fails to output visually correct result.

## C.2 FULL RESULT OF THE COMPARISON WITH BASELINES

We compare our method against three strong baselines on our proposed extensive dataset (Sec. 8.2). Fig. 9 shows the complete optimization loss of our method and the baselines, where we observe that all these methods give similar and sufficiently small loss except for the failure cases. Fig. 11 shows the volumetric density difference between the optimized result and ground truth, Fig. 12 shows the difference of the result on some unseen test sets, and Fig. 13 shows the difference of center of mass. In these three figures, we all observe that our method achieve the most accurate result. Fig. 10 shows the complete volumetric generation time, where we can observe that our methods is faster than the baselines in all examples. Tab. 3 lists the complete reconstruction quality, where our method beats all baselines except the toy-1 example.

We visualize the simulation and optimization result in Fig. 14, Fig. 15, and Fig. 16. Note that in Fig. 14 and Fig. 15 we visualize the difference of optimized physical parameters, and our results are more uniform and natural than all baselines. In Fig. 16, we show respectively the initial state, final state, and the physical distribution of the rigid bodies, and we plot an additional vertical line to help with interpretation. With our optimization of the internal structure, an originally unstable object stands steadily with reasonable alignment to the vertical line.

## C.3 ADDITIONAL REAL-WORLD EXAMPLES

**Hawk** A balanced hawk (Fig. 17) is a toy with a meticulously designed hollowed structure inside so as to balance itself with support at its beak only (Fig. 17 left). Similar to the rigid experiments in synthetic examples, the loss is defined as the distance between $c$ and the support center (the bird beak location).

Fig. 17 summarizes the results of our pipeline on this task. We begin with an initial guess of a solid hawk stuffed with homogeneous materials, which easily falls off from its support location (Fig. 17 top middle). After optimization, our pipeline deduces a hollowed design (Fig. 17 right) that successfully reproduces a convincing balanced hawk, verified by simulating its dynamic motion and observing negligible derivation or no tendency of falling off (Fig. 17 bottom middle).

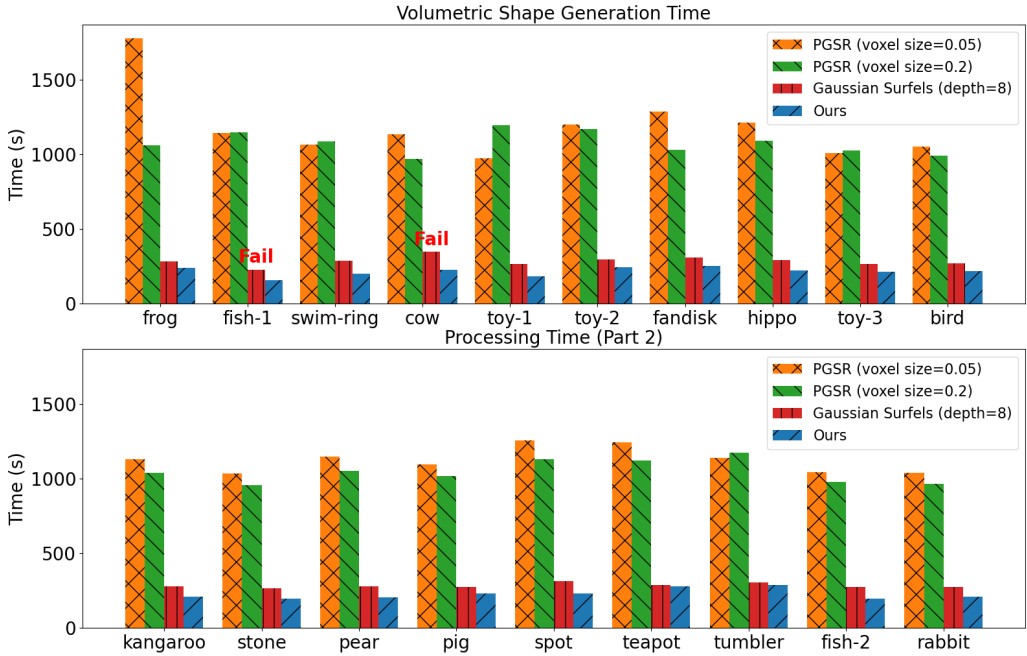

Figure 10: Full data of the volumetric shape generation time of our method and the baselines, where "fail" on the top means that the method fails to output visually correct result.

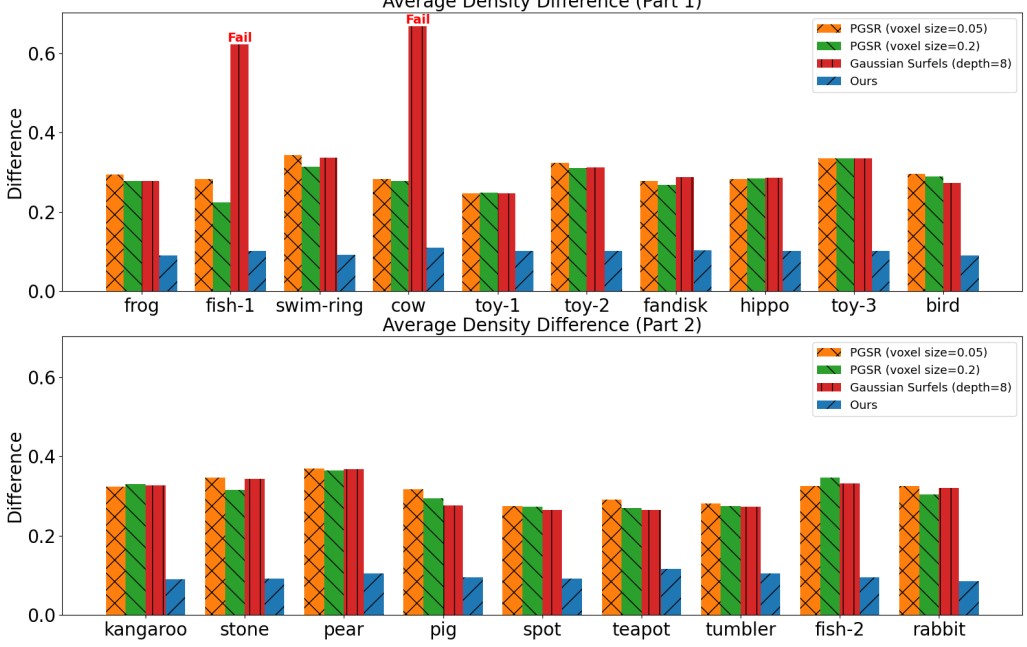

Figure 11: Full data of the density difference of our method and the baselines, where "fail" on the top means that the method fails to output visually correct result.

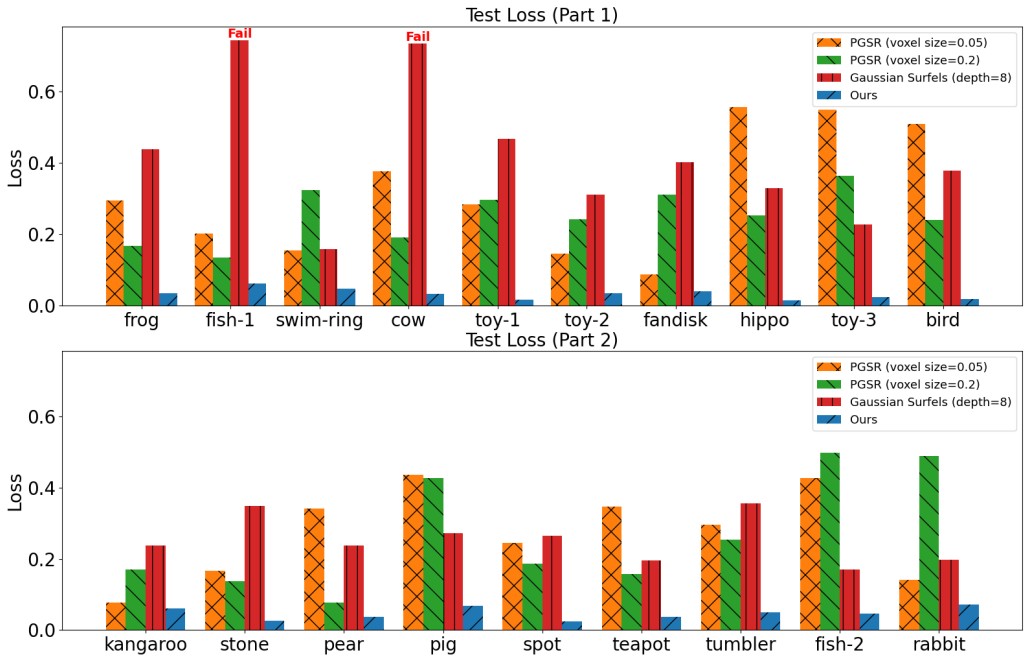

Figure 12: Full data of the test set loss of our method and the baselines, where "fail" on the top means that the method fails to output visually correct result.

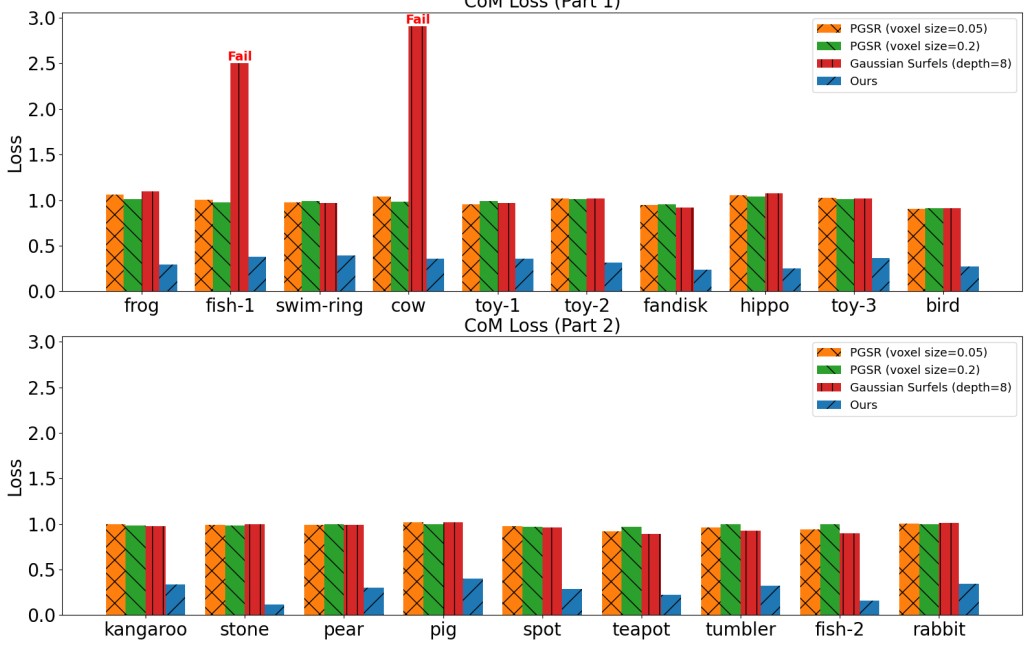

Figure 13: Full data of the center of mass (CoM) loss of our method and the baselines, where "fail" on the top means that the method fails to output visually correct result.

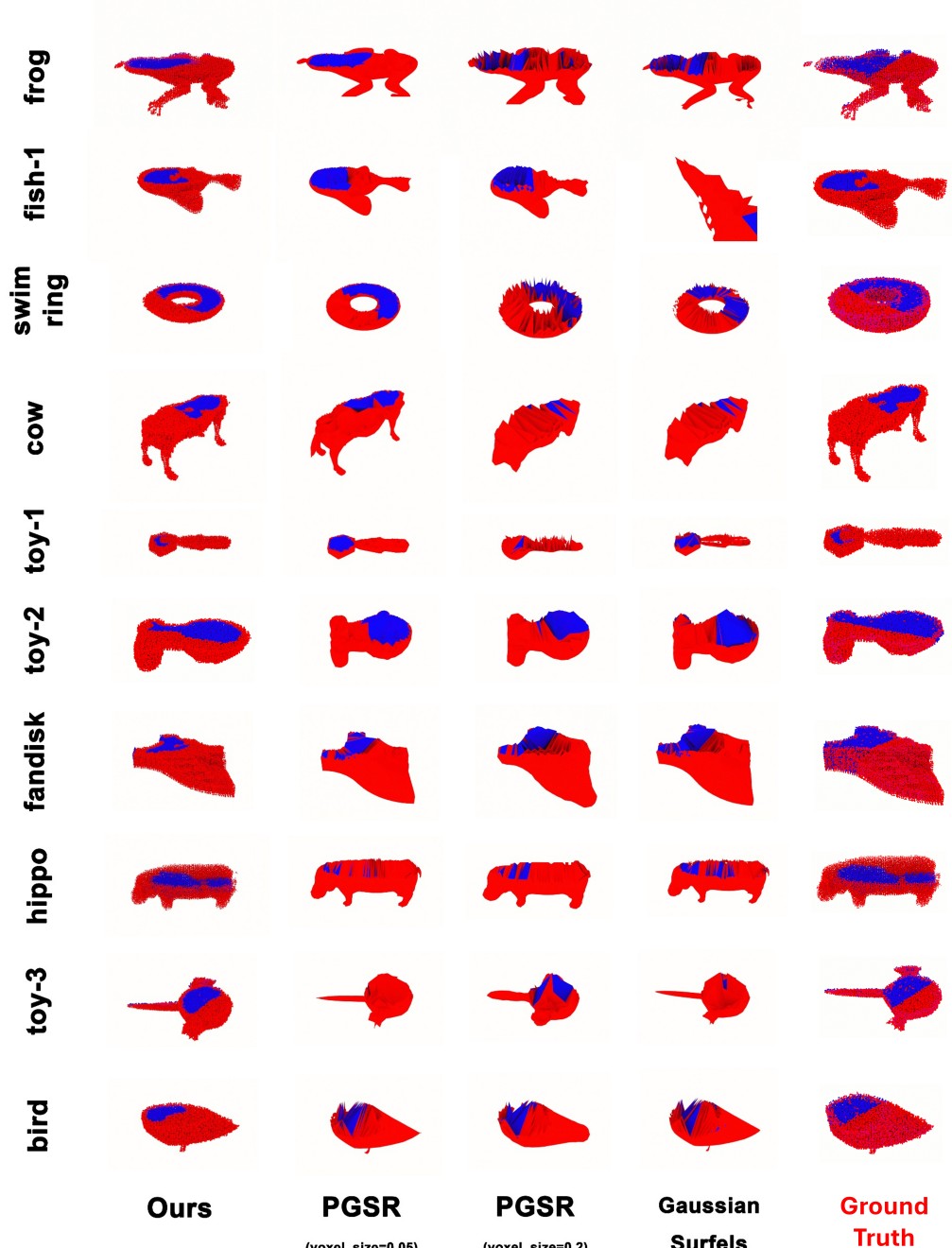

Figure 14: Full result (part 1) of the comparison of slices between our methods and baselines (Sec. 8.1)

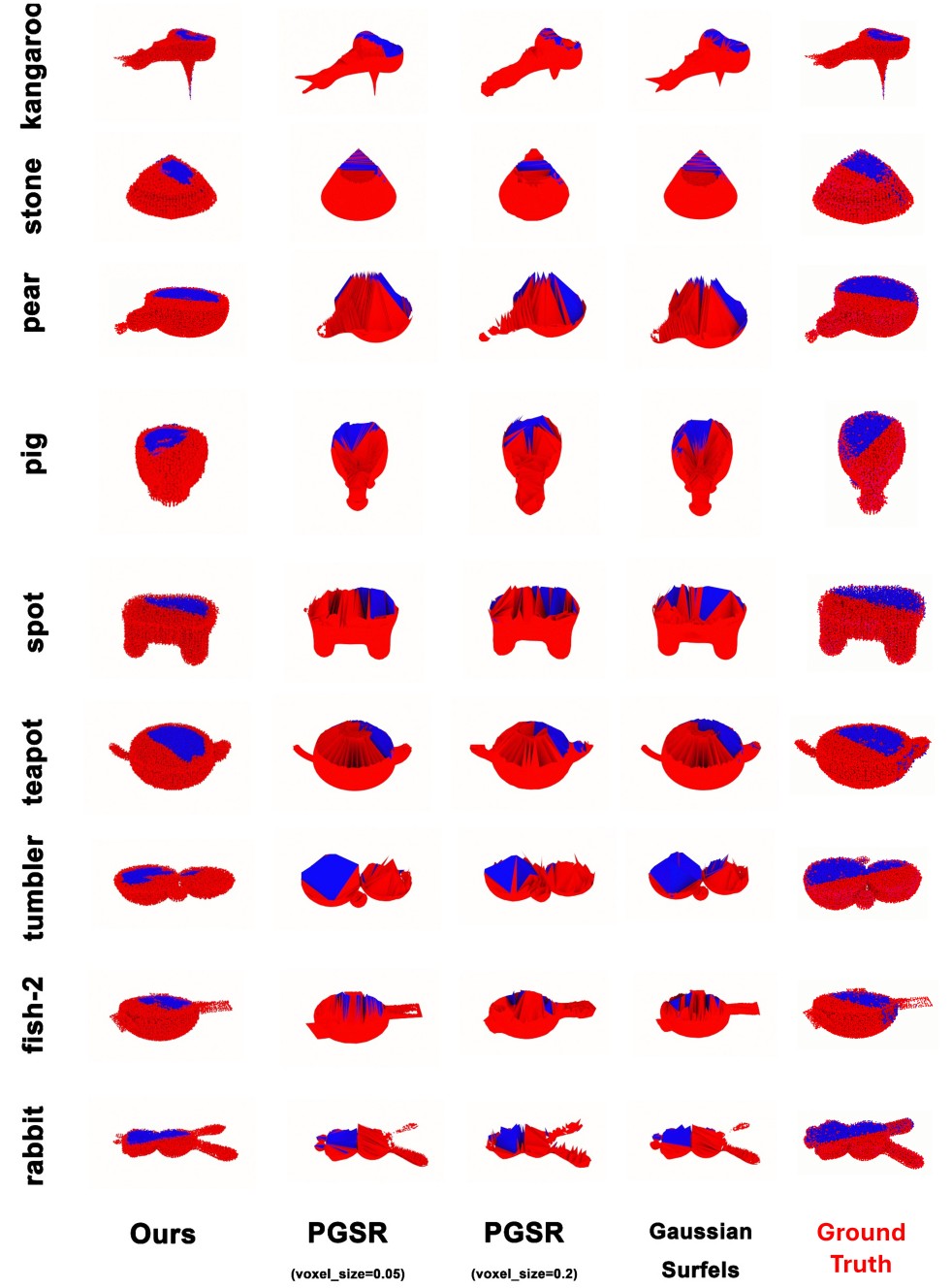

Figure 15: Full result (part 2) of the comparison of slices between our methods and baselines (Sec. 8.1)

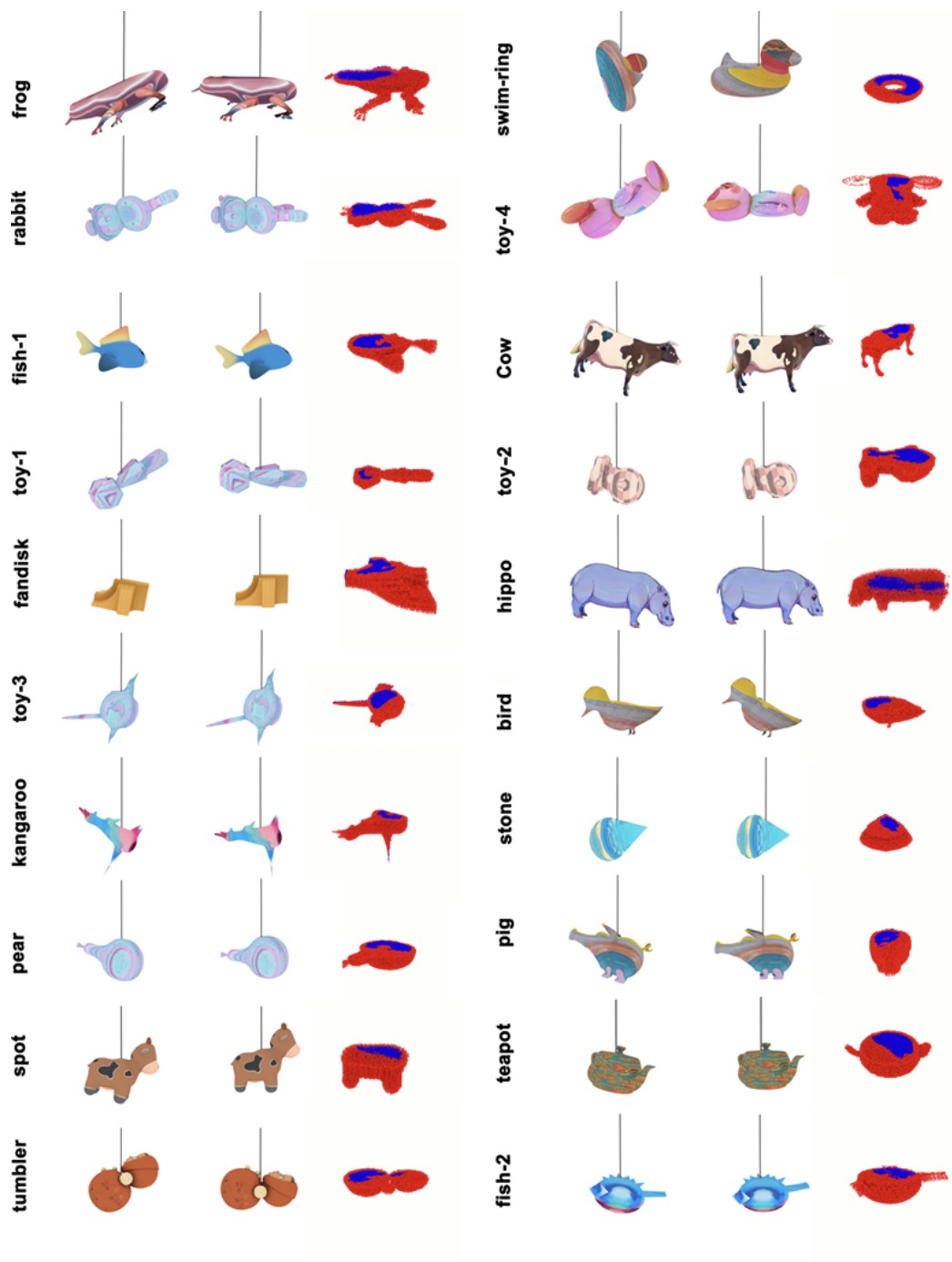

Figure 16: Full result (part 2) of the simulation of initial and final poses of the objects (Sec. 8.1)

Table 3: *Reconstruction quality.* Lower numbers indicate better performance. The optimal results are **bolded**.

| Dataset id | frog | fish-1 | swim-ring | cow | toy-1 | toy-2 | fandisk |
|---|---|---|---|---|---|---|---|
| Ours | **0.306** | **0.358** | **0.242** | **0.300** | 0.294 | **0.274** | **0.335** |
| PGSR (voxel size=0.05) | 0.745 | 0.998 | 0.464 | 0.410 | **0.263** | 0.761 | 0.904 |
| PGSR (voxel size=0.2) | 0.781 | 0.907 | 0.894 | 0.801 | 0.595 | 0.845 | 0.878 |
| Gaussian Surfels | 0.851 | Fail | 0.882 | Fail | 0.552 | 0.939 | 0.865 |

| Datset id | hippo | toy-3 | bird | kangaroo | stone | pear | pig |
|---|---|---|---|---|---|---|---|
| Ours | **0.663** | **0.285** | **0.268** | **0.388** | **0.290** | **0.245** | **0.343** |
| PGSR (voxel size=0.05) | 0.910 | 0.478 | 0.846 | 0.841 | 0.907 | 1.078 | 0.891 |
| PGSR (voxel size=0.2) | 0.869 | 0.720 | 0.850 | 0.480 | 0.904 | 1.027 | 0.867 |
| Gaussian Surfels | 0.939 | 0.459 | 0.906 | 0.976 | 0.890 | 1.148 | 0.939 |

| Dataset id | spot | teapot | tumbler | fish-2 | rabbit |  |  |
|---|---|---|---|---|---|---|---|
| Ours | **0.347** | **0.264** | **0.305** | **0.352** | **0.298** |  |  |
| PGSR (voxel size=0.05) | 0.822 | 0.822 | 0.953 | 0.605 | 0.930 |  |  |
| PGSR (voxel size=0.2) | 0.901 | 0.841 | 0.755 | 0.889 | 0.903 |  |  |
| Gaussian Surfels | 0.724 | 0.850 | 0.963 | 0.472 | 0.940 |  |  |

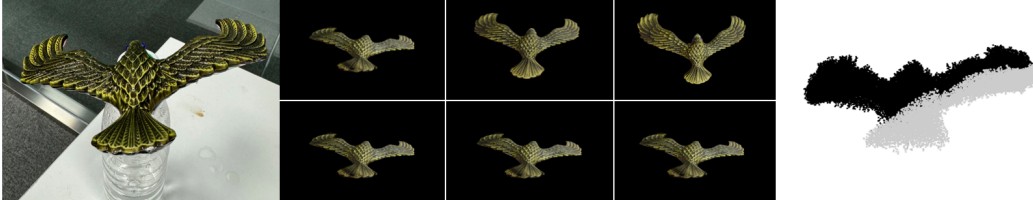

Figure 17: *Hawk* (Sec. 8.3). Left: a real-world photo of the balanced hawk resting on a bottle at its beak; Top middle: the falling-off motion from simulating the hawk point cloud with homogeneously stuffed materials; Bottom middle: the balanced motion from simulating the same hawk point cloud with optimized hollowed structure; Right: the optimized structure with black and gray particles indicating the solid and empty region, respectively. Note that the boundary particles are fixed as solid during optimization and removed from this figure for better visualization of the interior structure.

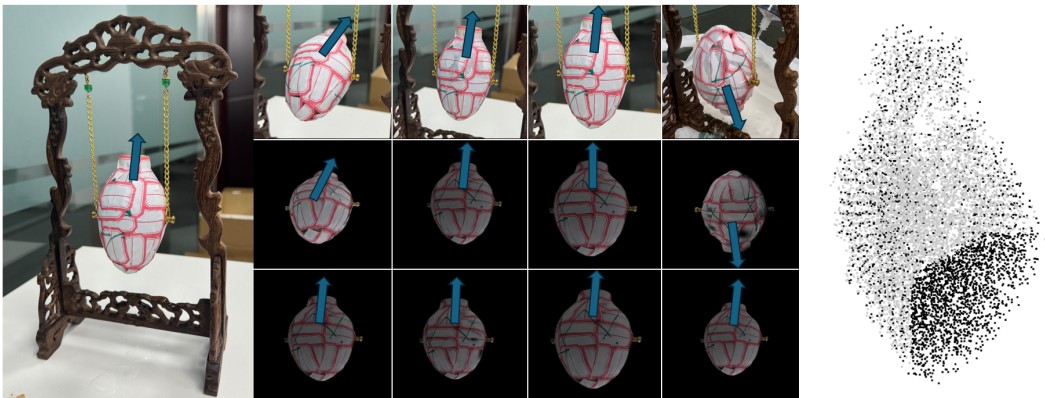

Figure 18: *Qiqi* (Sec. 8.3). Left: a photo of qiqi showing a tilting angle. The blue arrow indicates the up direction of qiqi. Middle: the pose changes of qiqi when increasing its capacity from empty to maximal; Middle: simulated pose changes of a qiqi with the same exterior from Gaussian splatting and an interior structure optimized from our pipeline; Bottom: simulated motion of a qiqi with an initial guess of uniformed hollowed interior. Right: the optimized interior structure by our pipeline.

**Qiqi (Tilting Vessel)**   Qiqi is a tilting vessel originating from ancient China (Fig. 18 left). It has a carefully crafted interior shape that exhibits representative poses at three different states: it tilts by a specific angle when empty; the tilting angle gradually decreases to zero when the vessel becomes half full; the vessel overturns (upside down) when reaching its maximal capacity. Fig. 18 top middle demonstrates this behavior of qiqi with representative frames from a video recording of filling water into qiqi. The blue arrow in the figure indicates its up direction. Note how the up direction is eventually upside down when qiqi reaches its capacity. Our goal is to infer an interior geometry of qiqi that reproduces all three representative poses.

We assign one objective for each representative pose of qiqi. The objective consists of two terms characterizing the preferred height and tilting angle from the center of mass. Note that for a given shape parameter $s$, the three poses compute three centers of mass based on considering the union of the shell characterized by $s$ and its stuffed interior at three specified height. This initial guess (a shell with uniform thickness) leads to a design resilient to changing tilting angles (Fig. 18 bottom middle), which shows that an interior geometry that can exhibit desired behaviors is not designed trivially.

Fig. 18 right reports our optimized interior shape of qiqi. The result clearly shows that an asymmetric design is necessary and expected due to the initial nonzero tilting angle when the vessel is empty. We simulate our inferred design with an increasing capacity and observe similar pose changes to the video recording (Fig. 18 middle), confirming the efficacy of our optimization pipeline.

## C.4   ABLATION STUDIES

**The density of points**   In order to investigate the impact of the density of points to our method, we thoroughly sample five different initialization: 12226, 20480, 29160, 40000, and 69120, and we show the result of this ablation study in Fig. 19. In spite of minor variances on the top of the wobble doll, the general quality of distribution decreases marginally, indicating the efficiency and the robustness of our method in terms of the number of initial points.

**The material property and object size**   To study how the choice of the size and the material of 3D object impacts the result of optimization, we design an ablation study by doubling the initial size and the material density of the hawk respectively. We show the results in Fig. 20, where the left figure is the original result, the middle the doubled size, and the right the doubled material density. As shown in the figure, the optimization results look very similar to each other, indicating that our method is not sensitive to the choice of material and the overall scale.

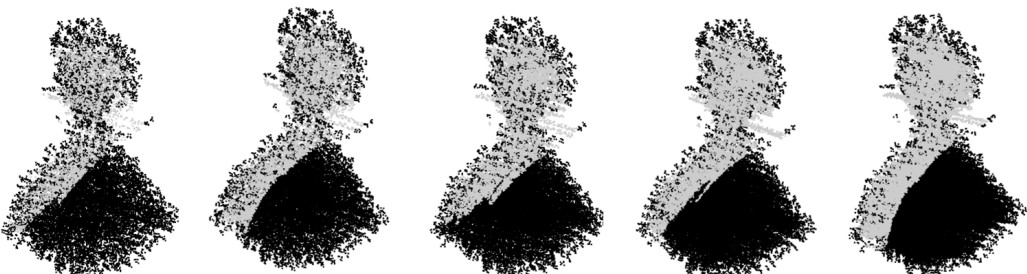

Figure 19: *The density of points.* Black points represent solid part and grey points represent hollow part. From left to right, the density of points are 12226, 20480, 29160, 40000, 69120 respectively. We can observe that the quality of the result does not decrease much when the density of points decreases.

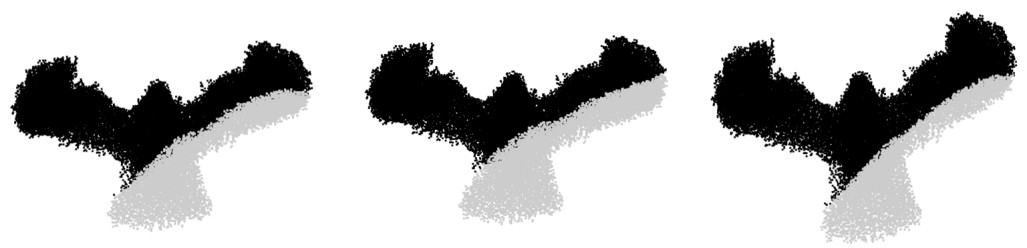

Figure 20: Ablation study on the material and size. Black points represent solid part and grey points represent hollow part. The first graph is the original setting. The second graph is doubling the scale of the hawk. The third graph is doubling the material density of the hawk. We can observe that with the same relative shape and relative physical parameters, the output does not change with its absolute value.

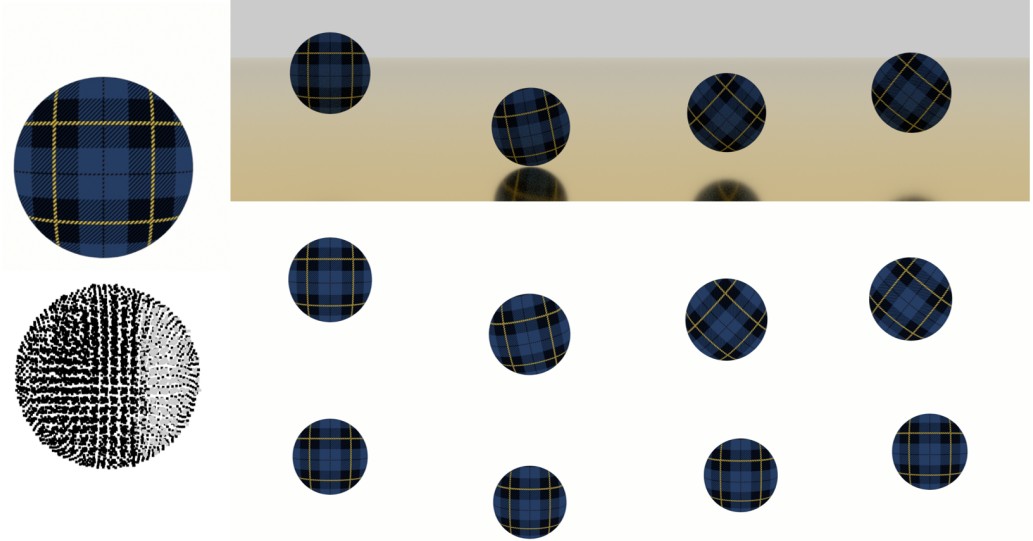

Figure 21: *Rigid-body simulation with collision*. The top left part demonstrates the appearance of the ball. The four pictures in top right part is four frames from the input video (ground truth). The four pictures in middle right part is the corresponding frames of the final solution, and the four pictures in bottom right part is that of the initial guess. The bottom left part is the internal structure, where black points represent solid part and grey points represents hollow part.

**The type of simulation**   To show the extensibility of our method to more types of physical phenomena, we combine the rigid-body physical simulation with the proposed collision handling and apply our method on it. As shown in Fig. 21, we drop the rigid ball on the floor and record the ground-truth trajectory as shown on the top row. After the optimization of interior structure, the output motion trajectory matches the visual observation. We want to highlight that in the original video the rigid ball was rotating, which is quite hard for traditional methods to capture. Our method, however, reconstructs the behavior nicely as indicated by the textural rotation.

**The lighting condition and camera pose**   In addition to the real-world wobbly doll, we create a simulated wobbly doll in our synthetic environment as well for better reproducibility. We vary the lighting condition and the camera pose in order to test the robustness of our method as shown in Fig. 22. We did three experiments: (left) the normal light with front pose, (mid) the dark light with front pose, and (right) the normal light with side pose. We also attach the corresponding internal solution on the side. As indicated by the solutions, our method is robust in terms of varying lighting conditions and camera poses. We also argue that changing camera pose is equivalent to transforming the local frame of the dynamics, and our method is insensitive to both the changes.

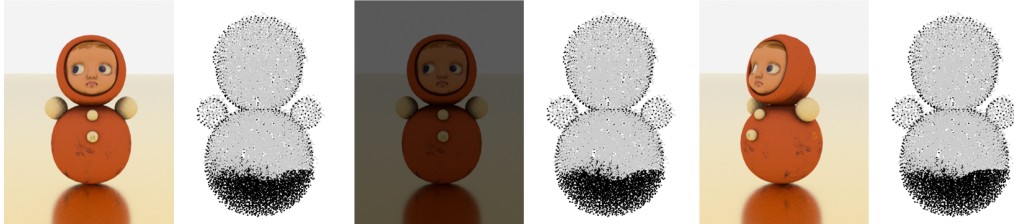

Figure 22: *Synthetic wobbly doll.* The experiment with the same setting of the real world wobbly doll but performed on the synthetic one in our dataset. Besides, we change its lighting and view point. Black points represent solid part and grey points represent hollow part. The first graph is bright lighting with front view, and the second graph is its solution. The third graph is dark lighting with front view, and the fourth graph is its solution. The fifth graph is bright lighting with side view, and the sixth graph is its solution. We can observe that the output of our pipeline does not change with the lighting and camera position.

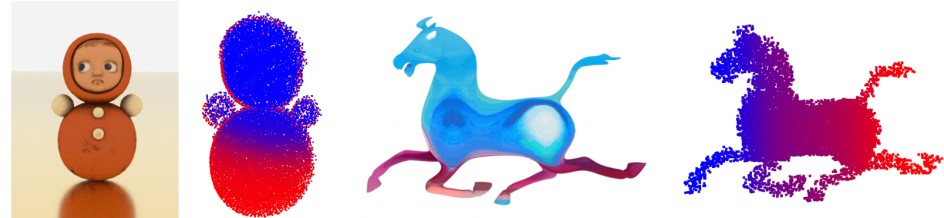

Figure 23: Experiments which supports continuously varying material. The two experiments share similar setting with the previous wobble doll and horse experiment, while the main difference is we allow a continuously varying material. The first and third figures are two typical input images, while the second and fourth figures are the optimized result. The color indicates the density of each point, where red means higher density (solid) and blue means lower density (hollow).

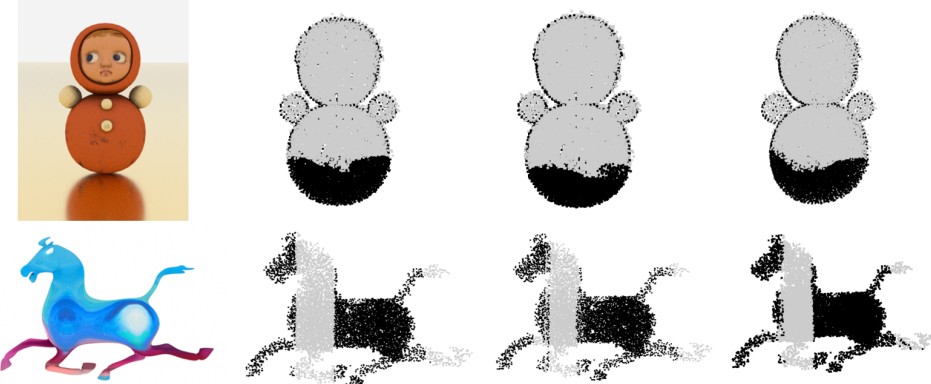

Figure 24: Experiments using multi-video for optimization. The two experiments share similar setting with the previous wobble doll and horse experiment, while we leverage multi-video optimization, with videos of difference view points, light conditions, vibration amplitudes, etc. From left to right, the first figure represents the rendered object, the second represents the ground truth, the third represents the single-video result, the fourth represents the multi-video result.

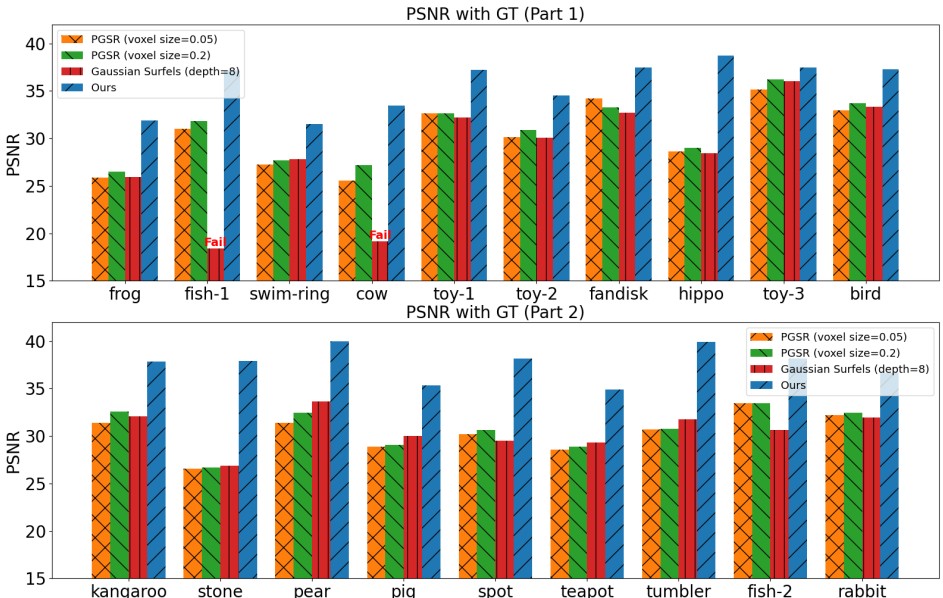

Figure 25: Full data of the PGSR of rendered result with respect to the ground truth, where "fail" on the top means that the method fails to output visually correct result.

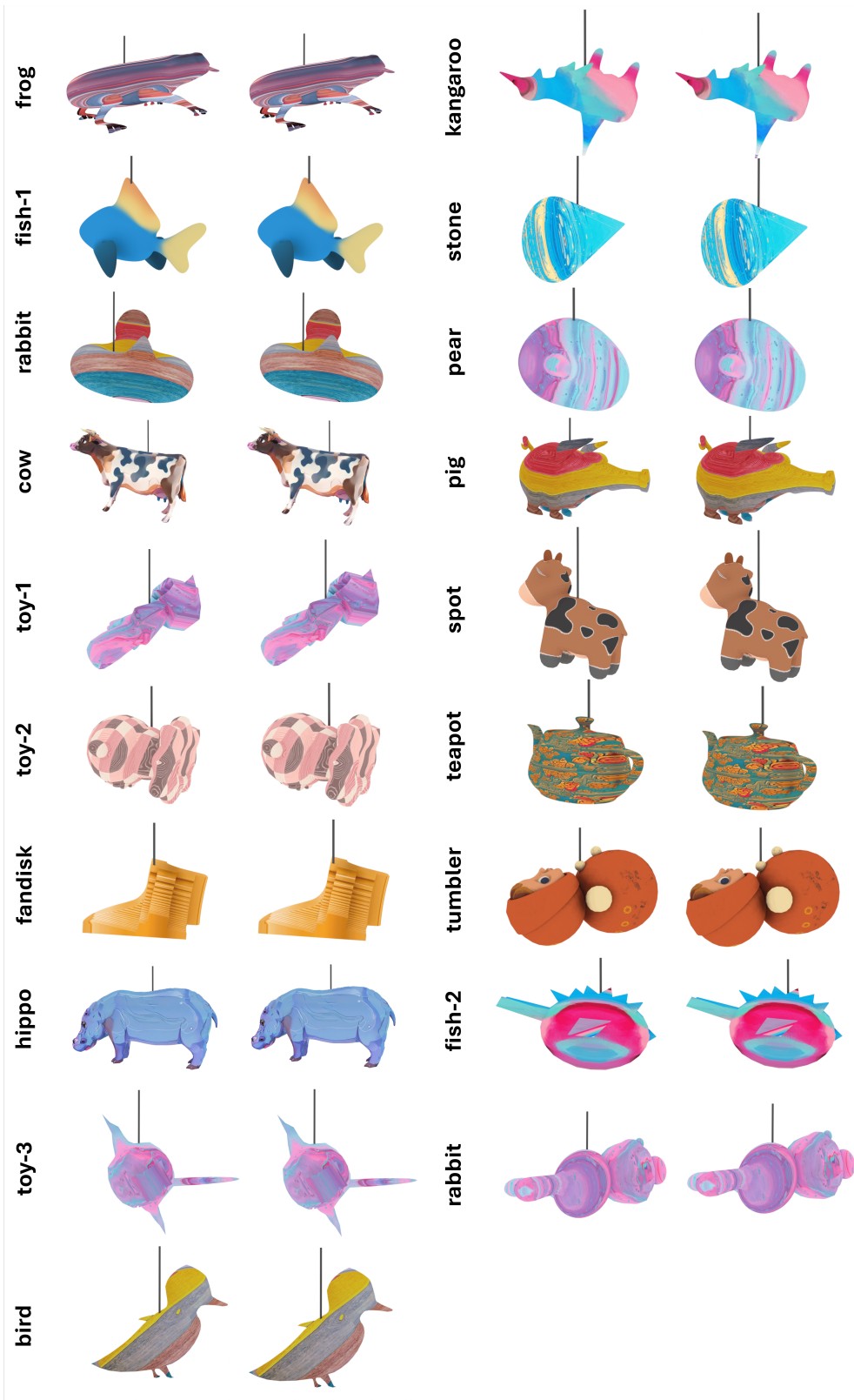

Figure 26: One example of the comparison in the unseen test set. The left figure represents the ground truth and the right figure represents our simulated result.

