# OpenReview forum: "TopoGaussian: Inferring Internal Topology Structures from Visual Clues"
_ICLR.cc/2025/Conference — ICLR 2025 Poster_

### Official Review · Reviewer_HpwZ · 2024-10-31

**Soundness:** 3
**Presentation:** 3
**Contribution:** 3
**Rating:** 6
**Confidence:** 3

**Summary:**

This paper proposed a particle-based pipeline for motion reconstruction with physically corrected internal topology based on visual inputs. Specifically, they used a vision-based reconstruction method to construct the point cloud and optimize the physical properties of each particle through a differentiable simulation. Through several rigid-body and soft-body motion tasks, they verified the validity of the approach.

**Strengths:**

● Extracting an object's internal topology through a visual solution sounds interesting and is a novel task. The feasibility of the solution is also demonstrated.

● The article constructs multiple synthetic data to validate the effectiveness of its scheme, and more importantly, it accomplishes multiple sets of validation experiments in real-world scenarios.

**Weaknesses:**

● The main goal of the article is to build the internal structure of the object by visual motion, but there is no proper metric or GT comparison to measure the correctness of the internal structure recovery, it only compares the rendering results

● The comparative experiments are insufficient; the paper only compares optimization loss, reconstruction quality, and time with two mesh-based approaches. This fails to highlight the main contribution of the paper, which is the recovery of internal structures. Additionally, the visual comparison of internal structures does not fully reflect the accuracy of the internal structure recovery.

● The proposed method involves multiple steps, such as Gaussian Splatting, Volumetric Shape Generation, and Topology Optimization, etc., but lacks thorough validation for these steps. For example, there is no detailed analysis of the impact of filling quality on topology optimization or the robustness of the entire system to the visual input quality and movement amplitude.

**Questions:**

● Is there a unique solution for recovering the internal structure of an object based solely on visual observation? How can we determine if the recovered internal structure is reasonable?

● The article mentions that the volumetric shape generation stage is faster than mesh-based methods. Is this stage entirely consistent with the PhyGaussian?

● What is the relationship between the metrics used in the article—reconstruction quality and optimization loss? How is reconstruction quality defined? Does the "Smoothness" mentioned in Section 8.1 Metrics represent reconstruction quality? Why does the proposed method achieve similar optimization loss to the baseline, yet show significantly better reconstruction quality than the baseline?

---

> ### Author Response · Authors · 2024-11-22
> **Thank you for your review!**
>
> We appreciate the valuable comments from the reviewer.
>
> **1. (W1 & W2 & Q1) Reconstruction error with respect to ground truth structure**
>
> Thank you for pointing this out. We perform more quantitative analysis on the accuracy of reconstructed topology compared with GT with 3 metrics, and achieve **3.21x**, **7.51x**, and **3.47x** smaller error than the original baselines on average (Fig.11-13). Since this is a common question among all reviewers, we explain it with more details in the general response.
>
> **2. (W3) The proposed method involves multiple steps, such as ..., but lacks thorough validation for these steps.**
>
> Thank you for bringing this issue. The reviewer points out two important point to be tested in our pipeline:
> - Filling quality
> - Input motion quality
>
> We have the following ablation studies to test the robustness of these parts in our original paper:
> - In Fig.19, we test the influence of different filling resolutions, where we find that the results are not affected by different resolutions and output similar results
> - In Fig.22, we test the impact of different video quality, including light condition and view point, where the results also remain similar, proving the robustness of our pipeline.
>
> Moreover, we have performed another ablation study in Fig.20 to test the robustness with different material properties and object size, which also strengthen the robustness of our pipeline
>
> **3. (Q2)  The article mentions that the volumetric shape generation stage is faster than mesh-based methods. Is this stage entirely consistent with the Phy(s)Gaussian?**
>
> Thank you for asking this valuable question. We do not find ''PhyGaussian'' on the Internet and guess the reviewer means ''PhysGaussian''. The point cloud filling process (which transfers the surface point cloud from GS to a volume point cloud) in our pipeline is similar to PhysGaussian. On the other hand, since PhysGaussian does not contain topology optimization, the topology representation part, including the point and surface representation (neural implicit surface and quadratic surface) is not relevant to it.
>
> **4. (Q3) What is the relationship between the metrics used in the article—reconstruction quality and optimization loss?**
>
> Thank you for bringing this issue. We would like to discuss these two metrics together with the three new metrics (density loss, test loss, and CoM loss) we provide in the general response.
> There are many metrics we need to consider to measure the result:
> 1. **Motion loss of input** measures the difference between simulated motions of results and input.
> 2. **Motion loss of test set (unseen motions)** measures the difference of simulated motions between results and test set.
> 3. **Density loss compared with GT** measures the difference of output topology (density distribution).
> 4. **CoM loss compared with GT** measures the deviation of the center of mass.
> 5. **Smoothness loss of the result (3D printing concerns)** measures the average laplacian of the internal surface in output topology.
>
> We use the second to fourth metrics for model evaluation by comparing them with the ground truth (GT) in the testing set. These metrics are strictly reserved for evaluation and should not be utilized during optimization. On the other hand, the differential of the fifth loss is difficult to solve, which makes it hard to be added in the gradient-based optimization process. Therefore, we choose our optimization objective to minimize the first loss function, and use the other four metrics to validate and measure the accuracy and quality of the results. It may be a promising task to add the fifth loss as a regularization term in the optimization process and enhance the quality of the results in the future.

---

> > ### Author Response · Authors · 2024-11-29
> >
> > Dear reviewer HpwZ,
> >
> > We are truly grateful for your insightful comments and advice, which have played a significant role in enhancing the quality and clarity of our paper.
> >
> > We hope that the additional details and experimental results we provided have effectively addressed your concerns. As the rebuttal period comes to an end, we kindly request your thoughts on our rebuttal and ask that you consider raising your score accordingly. If there are any remaining concerns, please feel free to share them with us.
> >
> > Once again, we deeply appreciate your thoughtful review and constructive feedback.
> >
> > Best,
> >
> > Authors

---

> > > ### Comment · Reviewer_HpwZ · 2024-12-01
> > >
> > > Dear Authors:
> > > Thank you for your detailed explanations and additional experiments. After reading other reviews and your feedback, I think you have answered most of my questions and addressed my concerns, especially on the structure evaluation. I still maintain that this is an interesting and novel work. I am pleased to raise my score.

---

> > > > ### Author Response · Authors · 2024-12-01
> > > >
> > > > Thank you for your insightful and constructive feedback on our manuscript. We are encouraged by your positive comments on our approach, which motivate us to further explore this research area.
> > > >
> > > > We noticed that the score (5) in your original review hasn't been updated yet. Would it be possible for us to get your new score on our submission?
> > > >
> > > > If you have any additional suggestions or comments, please feel free to share them with us. Thank you once again for your time and thoughtful review!

---

### Official Review · Reviewer_2FAv · 2024-11-03

**Soundness:** 3
**Presentation:** 3
**Contribution:** 3
**Rating:** 6
**Confidence:** 4

**Summary:**

The paper introduces TopoGaussian, a particle-based approach that uses visual clues from photos and videos to infer internal topology structures of opaque objects. Key contributions include:
1. Mesh-Free Topology Inference: TopoGaussian combines Gaussian splatting and a particle-based differentiable simulator to infer interior topology without requiring mesh-based representations, which traditionally entail extensive fixing and filling processes.
2. Flexible Topology Representations: The pipeline supports different topology representations (particle, neural implicit surface, and quadratic surface), allowing for optimization and simulation within a unified framework.
3. Experimental Validation: TopoGaussian is evaluated on synthetic and real-world tasks, showcasing its capability to generate 3D-printable reconstructions that exhibit high fidelity and reduced processing time compared to existing mesh-based methods.

**Strengths:**

1. Efficient and High-Quality Reconstruction: The particle-based approach of TopoGaussian achieves efficient reconstructions, with the authors reporting a significant speedup (5.26x faster) and superior boundary reconstruction quality (2.33x improvement) compared to mesh-based methods like PGSR and Gaussian Surfels.
2. Annotation-Free and Flexible: The pipeline’s independence from intrusive sensors or annotation requirements makes it practical and applicable in fields like robotics and manufacturing. Additionally, the three topology representation options offer flexibility based on application needs, from rigid to soft-body simulation.
3. Simplicity and Smoothness of Output: By eliminating the need for mesh processing, TopoGaussian produces smoother outputs conducive to 3D printing and manufacturing.

**Weaknesses:**

1. Evaluation Limitation: The method optimizes the interior structure based on a single motion, which may lead to overfitting and an inaccurate reconstruction of the true internal structure. Since ground truth data for internal topology is unavailable, it is difficult to verify if the inferred structure is correct or merely adapted to the given motion. Although the authors acknowledge this limitation and propose alternative metrics, these may not fully reflect the true structure. To strengthen the evaluation, the authors could consider obtaining ground truth through simulation (maybe with physics simulation) or testing the inferred structure on new motion videos as a test set. If the predicted structure is accurate, it should exhibit consistent behavior across these unseen motions, providing stronger validation.
2. Potential Overfitting to Single Motion: The current approach optimizes the internal structure based on a single motion video, which may lead to overfitting and limit the model’s generalization capability. To improve the robustness and accuracy of the inferred structure, the authors could consider optimizing based on multiple motion videos. By incorporating a variety of motions, the resulting model may better capture the true internal structure and provide more reliable and generalizable results.
3. Limitation to Simple Material Compositions: The current framework supports only single-object, dual-material compositions, which may limit applications involving complex, heterogeneous materials or multi-object interactions. Future work could focus on extending support to more intricate material compositions.

---

The authors have provided additional results to address my concerns.

**Questions:**

See weakness

**Details Of Ethics Concerns:**

/

---

> ### Author Response · Authors · 2024-11-22
> **Thank you for your review!**
>
> We are grateful for the reviewer’s insightful comments.
>
> **1. (W1) Reconstruction error with respect to ground truth structure**
>
> Thank you for pointing this out. We perform more quantitative analysis on the accuracy of reconstructed topology compared with GT with 3 metrics, and achieve **3.21x**, **7.51x**, and **3.47x** smaller error than the original baselines on average (Fig.11-13). Since this is a common question among all reviewers, we explain it with more details in the general response.
>
> **2. (W2) Multi-video training**
>
> This is a brilliant experiment proposal. We have performed new experiments on the synthetic wobble doll and horse examples to leverage this idea. The experiments share similar settings with the original ones, while we generate several different videos (with different view point, light conditions, vibration amplitude, etc.) to perform optimization, with optimization loss based on all the input motions. We can observe that our model successfully outputs a smooth, accurate result with multiple motions as input, which matches the physical characteristics properly, with almost identical final optimization loss. For visual demonstration, please refer to Fig.24 in appendix.
>
> **3. (W3) Limitation to simple material compositions**
>
> Thank you for bringing up this issue. We would like to clarify the ability of our pipeline in the two mentioned parts:
> - **Continuously varying material**: Our method choose to focus on a dual-material setting due to the application concerns. In practice, it is difficult to manufacture a continuously varying topology structure by traditional methods or 3D printing, which means that a continuous result will be unrealistic to implement in industry. Therefore, we add a sharp sigmoid function in our pipeline to restrict our output to an industrial-friendly dual material result.
> That said, if we choose a smoother sigmoid function, our pipeline can also optimize a continuous topology structure. We use an experiment on the synthetic wobble doll and horse to demonstrate this, with similar settings to the original ones. The final optimization loss is **2.2e-5**, which is similar to the binary material (**5.6%** difference), while the visualization is shown in Fig.23.
> - **Multi-object setting**: Our pipeline can handle the interaction between objects through the collision-handling system mentioned in Sec 6.3, and has been exhibited in the collision experiment in Fig.21. The limitation here is that although we allow interaction, we can only optimize the topology structure of a single object, which is an intriguing problem for future work.

---

> > ### Comment · Reviewer_2FAv · 2024-11-23
> > **follow-up questions**
> >
> > ## Comparison Against GT
> > Thank you for providing the additional experiments and quantitative comparison (density difference/unseen video) against the ground truth. This strengthens the evidence supporting the method's effectiveness.
> > I would also suggest including a qualitative comparison since ground truth is now available. For example, in Figures 14–15, adding an additional column with the ground truth would provide a clear visual reference, enhancing the interpretability of the results.
> >
> > ## Unseen video evaluation
> > 1) I suggest including a side-by-side comparison video for the test set. For example, a minimal demonstration could involve showing the ground truth test video on the left side and the rendered video on the right side.
> >
> > 2) Additionally, I recommend including commonly used metrics for image rendering evaluation, such as PSNR, SSIM, and LPIPS. These metrics, as referenced in the 3DGS paper, provide a more direct and standardized quality assessment compared to the current metric (optimization loss). These values would help readers better understand the rendering quality of the method.
> >
> > ## Multi-Video Training
> > Thank you for extending the method to include multi-video training and providing the corresponding results.
> > As the authors have incorporated this extension, is there a comparison between the results of multi-video training and single-video training, both quantitatively and qualitatively? Specifically, does multi-video supervision provide any measurable improvement in reconstruction accuracy or quality?
> > Currently, Figure 24 shows the results for multi-video training, but there is no reference provided. It would be beneficial to include the ground truth structure and the results from single-video training as a comparison. Without these references, the multi-video training experiment lacks sufficient context to demonstrate its additional merit to the paper.

---

> ### Author Response · Authors · 2024-11-24
> **Thank you for your reply!**
>
> Thank you for proposing the advice to strengthen our work. We have revised the manuscript based on the advice and will briefly explain the revision.
>
> **1. Comparison Against GT**
>
> Thank you for suggesting the qualitative comparison with the ground truth. We have added an extra column in Figs. 14 & 15 to demonstrate the ground truth structure. We observe that our method outputs the closest result, which is consistent with the quantitative results reflected in Fig.11.
>
> **2. Unseen video evaluation**
>
> Thank you for suggesting the additional statistics and qualitative comparison. We have added Fig.26 to visualize one example in our unseen test set, where we observe that the motion of our inferred structure is almost identical to that of the ground truth. We have calculated the PSNR value of our method and the baselines with respect to the ground truth and reported the statistics in the following table:
> | Method | Ours | PGSR (0.05 voxel size) | PGSR (0.2 voxel size) | Gaussian Surfels
> | -------- | -------- | -------- | -------- | -------- |
> | PSNR | **36.6** | 30.4 | 30.8| 29.5|
>
> where we can observe that our method achieves the highest PNSR, leading to the most accurate results. The full statistics are exhibited in Fig.25.
>
> **3. Multi-Video Training**
>
> Thank you for your advice on the further comparisons. We have updated Fig.24 to add comparison of the multi-video result against the single-video result and the ground truth, where we observe that the multi-video and single video results are generally similar. To analyze the new results quantitatively, we have also calculated their density loss compared with the ground truth and reported the statistics in the following table:
> |Example|Single-video|Multi-video|
> |------|------|------|
> |Wobble doll|0.1015|0.0968|
> |Horse|0.1001|0.0987|
>
> Here we can observe that the multi-video loss is slightly smaller than the single-video loss.

---

> > ### Comment · Reviewer_2FAv · 2024-11-25
> > **Thanks for the additional results**
> >
> > I would like to thank the authors for providing additional results, both quantitative and qualitative. This paper addresses an intriguing problem—internal structure reconstruction—which is not as extensively explored as external surface reconstruction. My initial concerns were primarily related to the evaluation protocols and the lack of sufficient results to demonstrate the method's effectiveness. However, the additional results provided by the authors have successfully addressed these concerns. As a result, I am pleased to raise my score.

---

> ### Author Response · Authors · 2024-11-26
> **Thank you for your review!**
>
> Thank you for your constructive review on the additional experiments! Your feedback has helped us a lot to improve our manuscript. We are also encouraged by your comment on the interior structure reconstruction problem we study, and we hope our work can attract more researchers to investigate this less explored problem and inspire interesting follow-up applications.
>
> If you have any other comments or suggestions on improving our work, please feel free to let us know. Thank you again for your time and review!

---

### Official Review · Reviewer_q35E · 2024-11-03

**Soundness:** 2
**Presentation:** 3
**Contribution:** 3
**Rating:** 6
**Confidence:** 4

**Summary:**

The paper presents TopoGaussian, a pipeline for inferring the internal structure of opaque objects using only photos and videos as input. The pipeline works by first using Gaussian splatting on multi-view images to obtain a point cloud, then optimizing the internal topology structure through a differentiable physics simulator to match observed motion patterns.

The key contributions include a particle-based, mesh-free pipeline that combines Gaussian splatting with a differentiable physics simulator; three flexible topology representation options: particle-based, neural implicit surface, and quadratic surface; a particle-based differentiable simulator supporting both rigid and soft objects with different topology structures.

**Strengths:**

1. Novel combination of Gaussian splatting with physics-based optimization for internal structure inference.
2. Uses a mesh-free approach that avoids common issues with mesh processing; presents particle-based differentiable simulations that are compatible with three flexible topology representations, including particle, neural implicit surface, and quadratic surface.
3. Well-structured presentation with a clear pipeline overview.

**Weaknesses:**

The reviewer appreciates the authors' effort in building a particle-based pipeline to find a physically plausible internal topology structure. As the authors have also mentioned in the paper, this task is relatively new, and there are fewer baselines to compare with (at least other baselines do not use point-based representation). I have several concerns about the measuring metrics and their validation to support the claims from the authors:

1. Optimization Loss: This measures the difference between simulated motion and reference motion, and it directly indicates whether the internal topology structure is physically plausible. However, in Figure 3, the current method does not achieve the lowest loss among baselines in multiple test samples.
2. Comparison Implementation: When exporting the mesh from other baselines and chaining it into the rest of the pipeline in this paper, how can the mesh-based representation be made compatible with the rest of the system that is particle-based?
3. Time / Smoothness: It is unclear whether this improvement comes from the GS representation itself or from the authors' method. The reviewer encourages the authors to elaborate more on this or provide ablation studies to explain that the improvement comes from the proposed method itself.
4. Inner Structure: Can the authors provide the reference ground truth for the inner structures when making comparisons with other baselines? The reviewer understands that, in practice, the inner structures are hard to acquire, but in synthetic data, it is practical to obtain the ground truth of inner structures.

Other question:

How is the decision variable applied to the point cloud representation to obtain a continuous indicator function from the point cloud? (Line. 193)

**Questions:**

Please see my questions in the previous weakness section.

---

> ### Author Response · Authors · 2024-11-22
> **Thank you for your review!**
>
> We appreciate the reviewer's effort in reading and evaluating our work carefully!
>
> **1. (W1&W4) Optimization loss and comparison between the ground truth topology and inferred topology**
>
> Thank you for pointing this out. We would like to clarify that the optimization loss in Fig.3 is the residual during the optimization process (the L2 loss between the optimized motion and input motion). It is mainly used to show the two failure cases, mainly demonstrating the loss in the specific optimization case with only qualitative information.
>
> For more general and quantitative metrics to exhibit the effects of optimization, we provide 3 new metrics and perform experiments with our synthetic dataset, achieving **3.21x**, **7.51x**, and **3.47x** smaller error than the original baselines on average (Fig.11-13). Since this is a common question among all reviewers, we explain it with more details in the general response.
>
> **2. (W3) It is unclear whether the time/smoothness improvement comes from the GS representation itself or from the authors' method.**
>
> We thank the insightful question from the reviewer. We would like to clarify that all methods, including all baselines and our pipeline, is based on GS. All these methods require GS to extract the surface information from the input. The main difference between our method and the baselines lies in the further processing and representation of the information extracted by GS. In more details, the baselines in our paper build the simulator and topology optimizer based on mesh. On the other hand, our method builds a particle-based pipeline by introducing a particle-based differentiable simulator and topology representation (point, neural implicit surface, and quadratic surface), providing a more flexible and various topology representation. Therefore, the time/smoothness improvement comes from this part without relation to GS itself since GS is also used in baselines.
>
> **3. (W2) How can the mesh-based representation be made compatible with the rest of the system that is particle-based?**
>
> Thank you for bringing this issue. Mesh and particles are only two different geometry representations for the discretization in simulator and optimizer. We only need to perform a traditional particle resampling on the mesh to make it compatible with our pipeline. For example, in rigid cases, we perform a resampling on Eqn.1 and calculate the center of mass by $\mathbf c=\frac{1}{m}\sum \rho_iV_i\mathbf x_{ci}$.

---

### Official Review · Reviewer_vXS1 · 2024-11-03

**Soundness:** 3
**Presentation:** 2
**Contribution:** 3
**Rating:** 8
**Confidence:** 3

**Summary:**

The authors present a holistic approach to estimating the internal topology of objects from images. The work relies on the use of a particle-based differentiable simulator to estimate probable internal topology directly from the seen motion of an object. By this approach, the author is able to generate even real-world 3D printed versions with reasonable structure directly from a small video sequence.

**Strengths:**

The author works on a novel approach to solving a relatively novel problem for modern 3D Computer Vision. While the idea of reconstruction of internal topologies is not novel to the best of my knowledge, I have not seen much work trying to solve this for 3D Gaussian Splatting or NeRF-based approaches.
The work itself is decently written and shows a great evaluation to validate the quality of their methodology.
The authors can propose a holistic pipeline that should make their work easily usable for users, with the authors giving a significant amount of design possibilities.
In general, I am more than in favor of this work's results and core concept being interesting.

**Weaknesses:**

A large issue in this work is the convoluted writing. While still quite understandable, this work packs a significant amount of results, ideas, and concepts from many different fields.
As such, while many parts of the simulation (core contribution) have been well explained, much information regarding the volumetric representation is missing. As such, to improve the work (and make it complete), I would suggest the author add more information in the appendix.
Another larger issue is the motivation. While in Computer Graphics/Computer Vision, the challenge of estimating internal topologies is quite interesting, the estimation part might be a larger issue for practical, real-world use cases. In many real-world applications that rely on internal topologies, it is quite important that exact information is given, as this cannot assured by your model. I am still quite unsure when this work will be usable in real-world applications.

**Questions:**

- The authors claim one of the applications is in 3D printing; since I lack any knowledge of 3D printing, I am unsure about its weaknesses. But to enhance my understanding, why do we require the internal topology to be known for this? Shouldn't having the surface not be enough?
- Please fix the typo in line 175 "point clout"
- Please keep writing style consistent for example, line 234, "point-cloud"
- Would it be possible again to summarize for me what actually is the main goal and main application of this work?
- Not so much a question but rather a comment regarding Abstract Style (improvement/suggestion): Having this kind of structure usually improves understandability and readability: 1. What is the problem and why is it important? 2. What are the limitations of existing solutions? 3. What are the advantages of the proposed approach? 4. How does it work? 5. Summary of results

---

> ### Author Response · Authors · 2024-11-22
> **Thank you for your review!**
>
> We thank the reviewer for the constructive questions.
>
> **1. (W2) Reconstruction error with respect to ground truth structure**
>
> Thank you for pointing this out. We perform more quantitative analysis on the accuracy of reconstructed topology compared with GT with 3 metrics, and achieve **3.21x**, **7.51x**, and **3.47x** smaller error than the original baselines on average (Fig.11-13). Since this is a common question among all reviewers, we explain it with more details in the general response.
>
> **2. (Q4) Would it be possible again to summarize for me what actually is the main goal and main application of this work?**
>
> Thank you for bringing this issue. Our pipeline reads the photos and videos of an object as input, and outputs an interior structure which can explain the motion in the input. In many cases, simply guessing a fully solid structure from surface information will restrict the possibility of physical parameters. For example, if a wobble doll is fully solid, its center of mass will be restricted at a high position and it will be impossible to remain stable. Therefore, many practical applications require us to depict the internal topology in order to satisfy the physical characteristics.
> The application of our work covers many areas including computer vision, robotics and manufacturing. One example is to build a physical artifact from an online video of an unknown object, and use 3D printing to reconstruct it in the real world, even if the video is synthetic (AI generated). Another potential application is to validate the authenticity given a period of video by analyzing the interior structure of the objects in the video using our pipeline
>
> **3. (Q1) Why do we require the internal topology to be known for 3D printing?**
>
> Thank you for asking this question. In 3D printing, the printer prints the objects layer by layer, and needs to know the detailed structure of each layer. More concretely, we must tell the printer which part of the layer should be printed (solid part) and which should not (hollow part). This requires us to detailedly describe the interior topology of the object in our printing file.
>
> **4. (W1 \& Q2 \& Q3 \& Q5) Writing Problems including typos, writing styles and lack of information**
>
> Thank you for providing corrections and suggestions on our writing, and we have fixed those misses in our revised version. Besides, we have added more detailed information on our volumetric representation in appendix A.3, and revised the abstract based on the suggestions in Q5.

---

> > ### Author Response · Authors · 2024-11-29
> > **Look forward to post-rebuttal feedback**
> >
> > Dear reviewer  vXS1,
> >
> > We are truly grateful for your insightful comments and advice, which have played a significant role in enhancing the quality and clarity of our paper.
> >
> > We hope that the additional details and experimental results we provided have effectively addressed your concerns. As the rebuttal period comes to an end, we kindly request your thoughts on our rebuttal and ask that you consider raising your score accordingly. If there are any remaining concerns, please feel free to share them with us.
> >
> > Once again, we deeply appreciate your thoughtful review and constructive feedback.
> >
> > Best,
> >
> > Authors

---

> > > ### Author Response · Authors · 2024-12-03
> > >
> > > Dear Reviewer,
> > >
> > > Thank you for your constructive review on our manuscript. As we approach the end of the rebuttal phase, would it be possible for us to ask for your post-rebuttal feedback to help us enhance our work? Thank you.
> > >
> > > Best,
> > >
> > > Authors

---

### Author Response · Authors · 2024-11-22
**Authors' rebuttal: general questions and new experiments**

We thank all reviewers and the AC for their time and effort in reviewing and for insightful comments to strengthen our work. We update a revised version of the manuscript based on the suggestions from the reviewers. Besides the responses to individual reviewers, here we would like to highlight our contributions and new quantitative/qualitative results added in the rebuttal.

**1. Contributions:**
1. **[Motivation]** Our manuscript studies an important problem (3D internal topology reconstruction) [vXS1, HpwZ].
2. **[Method]** Our method is novel [vXS1, q35E, 2FAv] and provides a flexible pipeline for various applications [vXS1, q35E, 2FAv].
3. **[Experiments]** Our experiments cover multiple metrics with real world validation [vXS1, 2FAv, HpwZ], and our method outperforms the baselines [2FAv].


**2. New Results:**
1. **[Experiments: Reconstruction error with respect to ground truth structure]** We perform more quantitative analysis on the accuracy of reconstructed topology compared with GT with 3 metrics, and achieve **3.21x**, **7.51x**, and **3.47x** smaller error than the original baselines on average. Since this is a common question among all reviewers, we will explain it with more details in the following part and refer to Fig.11-13 for detailed statistics.
2. **[Extensions: Optimization on multiple videos]** Thanks for the proposal from reviewer 2FAv, we extend our pipeline to accept multiple videos as optimization input, and successfully output a result to satisfy the multiple physical characteristics with almost identical optimization loss compared to the single-motion result. (Fig.24).
3. **[Ablation Studies: Exhibition of continuous material]** Based on the proposal from reviewer 2FAv, we add experiments to exhibit the ability of our pipeline to output a continuously varying topology structure which can explain the motion in the input (Fig.23). The final optimization loss is **2.2e-5**, which is similar to the binary material (**5.6%** difference).

**3. Further explanation on reconstruction error with respect to ground truth structure:**
We thank all reviewers for asking this valuable question. We provide the following three metrics to measure the reconstruction error:
- Density loss: Based on the traditional practice, we defined the volumetric average density difference to depict the difference between optimized result and ground truth.
- Test loss: Based on the suggestion from reviewer 2FAv, we generate several **unseen** poses as a test set, and measures the difference between optimized result and ground truth based on this test set, which characterizes the generalization ability of our pipeline.
- CoM loss: Since the physical behavior of a rigid body is dominated by its center of mass (CoM) in rigid cases, we also test the difference of CoM between the result and ground truth.

Based on these 3 metrics, we perform experiments on the objects in our synthetic datasets with results shown in Fig.11-13 respectively. The following table summarize the average loss comparisons between our method and baselines:
|Baselines|Density Loss|Test Loss|CoM Loss|
|---------|------------|---------|--------|
|Ours|**0.099**|**0.038**|**0.301**|
|PGSR (0.05 voxel size)|0.307 (3.11x)|0.297 (7.46x)|0.990 (3.29x)|
|PGSR (0.2 voxel size)|0.295 (2.99x)|0.259 (6.52x)|0.988 (3.28x)|
|Gaussian Surfels|0.348 (3.53x)|0.342 (8.57x)|1.161 (3.86x)|

We can observe that our method outperforms all the original baselines in all three metrics. The reason may come from the particle-based pipeline providing a more flexible and various topology representation than mesh-based pipeline, leading to more accurate results and stronger generalization ability. This gives us a good motivation for a particle-based representation.

---

### Author Response · Authors · 2024-11-30
**Could AC help to reach out reviewers?**

Dear AC and SAC:

Thank you for efficiently handling our draft.

As we approach the end of the rebuttal phase, we noticed that post-rebuttal feedback from three borderline reviewers is still pending.

Could the AC assist in contacting these reviewers for their responses?

We greatly appreciate all your efforts!

Thank you,

Authors

---

### Author Response · Authors · 2024-12-04
**Rebuttal Summary**

We thank all reviewers and the AC for their time and effort in reviewing our work and providing insightful post-rebuttal feedback. Here we would like to summarize the highlights of our work and the discussion during the rebuttal period.

**1. Highlights:**

1. **[Motivation]** Our manuscript studies an **important** problem (3D internal topology reconstruction from 2D visual inputs) [vXS1, HpwZ].
2. **[Method]** Our method is **novel** [vXS1, q35E, 2FAv] and provides a **flexible** pipeline for **various** applications [vXS1, q35E, 2FAv].
3. **[Experiments]** Our experiments cover **multiple** metrics with **real world** validation [vXS1, 2FAv, HpwZ], and our method **outperforms** the baselines [2FAv].


**2. New Results:**
1. **[Experiments: Reconstruction error with respect to ground truth structure]** We introduced **four more metrics** to demonstrate the accuracy of our methods, and achieve **better** performance over the original baselines on all these metrics (Fig. 11, 12, 13, and 15), with visualization in Fig. 14, 15, and 26. This addresses the general suggestions from all reviewers for more metrics.
2. **[Extensions: Optimization on multiple videos]** We extend our pipeline to accept **multiple videos** as optimization input, and successfully output a result to satisfy the multiple physical characteristics with slightly better performance than single-video optimization (Fig. 24). This addresses the suggestion from reviewer 2FAv for the extension to multi-video training.
3. **[Ablation Studies: Exhibition of continuous material]** We add experiments to exhibit the ability of our pipeline to output a **continuously** varying topology structure which can explain the motion in the input (Fig. 23). This addresses the inquiry from reviewer 2FAv on the ability to process continuous material.

We hope our new experiments have addressed the questions raised in the reviews. We are encouraged by the unanimously positive scores from all reviewers after rebuttal. We thank all reviewers for the constructive discussions during the rebuttal phase and the time and efforts of AC to review our manuscript.

---

### Meta-Review · Area_Chair_4UPX · 2024-12-18

**Metareview:**

This paper introduces a novel method for reconstructing the internal topology structure of an opaque 3D object from a video and/or multi-view images. The pipeline first reconstructs the 3D object as a set of Gaussian splats. The internal topology structure is then represented using a solid part indicator for each point and an SDF for fine details. A differentiable particle-based simulator for dynamic motion, incorporating constitutive models, an actuation model, and a collision model, is employed to measure the L2 difference with the reference motion and perform gradient descent on the topology representation. The experiments demonstrate results with several real-world examples.

All reviewers gave positive scores and found that: the paper solves a novel and interesting problem; the method proposes a novel combination of Gaussian splatting and physics-based optimization; and the presentation is clear. While there were some concerns about the evaluation and validation of the design choices in each part of the method, most of them were addressed in the rebuttal. The discussion thus quickly converged to accept the submission.

**Additional Comments On Reviewer Discussion:**

There was no rebuttal. All reviewers were positive to accept this submission.

---

### Decision · Program_Chairs · 2025-01-22

Accept (Poster)